# WILD-Diffusion: A WDRO Inspired Training Method for Diffusion Models under Limited Data

**Xianglu Wang**[1], **Wanlin Zhang**[2,3], **Hu Ding**[2*]
[1]School of Artificial Intelligence and Data Science, University of Science and Technology of China
[2]School of Computer Science and Technology, University of Science and Technology of China
[3]Shanghai Innovation Institute
{wxlu, ideven, huding}@mail.ustc.edu.cn

## Abstract

Diffusion models have recently emerged as a powerful class of generative models and have achieved state-of-the-art performance in various image synthesis tasks. However, training diffusion models generally requires large amounts of data and suffer from *overfitting* when the dataset size is limited. To address these limitations, we propose a novel method called **WILD-Diffusion**, which is inspired by Wasserstein Distributionally Robust Optimization (WDRO), an important and elegant mathematical formulation from robust optimization area. Specifically, WILD-Diffusion utilizes WDRO to iteratively generate new training samples within a Wasserstein distance based uncertainty set centered at the limited data data distribution. This carefully designed method can progressively augment the training set throughout the training process and effectively overcome the obstacles caused by the limited data issue. Moreover, we establish the convergence guarantee for our algorithm even though the mixture of diffusion process and WDRO brings significant challenges to our analysis in theory. Finally, we conduct a set of experiments to verify the effectiveness of our proposed method. With WILD-Diffusion, we can achieve more than a $10\%$ reduction in FID using only $20\%$ of the training data across different datasets. Moreover, our method can attain state-of-the-art FID with as few as 100 images, both in pretrained and non-pretrained settings. The code is available at github repo.

## 1 Introduction

Diffusion models (Ho et al., 2020; Sohl-Dickstein et al., 2015; Song & Ermon, 2019; Song et al., 2021b) have become a leading family of deep generative models. Unlike generative adversarial networks (GANs) (Goodfellow et al., 2014) and variational autoencoders (VAEs) (Kingma et al., 2013; Rezende et al., 2014), which generate samples by decoding from a low dimensional latent variable, diffusion models learn to iteratively denoise a noise corrupted signal through a forward–reverse diffusion process (Yang et al., 2023b). Recent studies show that diffusion models have been shown to outperform GANs in many image generation tasks, including image editing (Huang et al., 2025b; Gu et al., 2023; Kawar et al., 2023; Yang et al., 2023a), image restoration (Xia et al., 2023; Fei et al., 2023; Zhu et al., 2023; Lin et al., 2024), style transfer (Zhang et al., 2023b; Wang et al., 2023d; Yang et al., 2023c), and text-to-image generation (Zhang et al., 2023a; Saharia et al., 2022; Ruiz et al., 2023).

However, the increasingly impressive results of diffusion models are fueled by the seemingly unlimited supply of images. In other words, diffusion models require large amounts of data for stable training (Wang et al., 2023a; Li et al., 2025; Zhang et al., 2025), which hinders the application of diffusion models in *limited data settings*. For example, training a vanilla diffusion model (Ho et al., 2020) on only 2,000 samples from the FFHQ dataset (Karras et al., 2019) (about $4\%$ of the full dataset) leads to a sharp performance drop, with the FID increasing from about 2.5 (full dataset) to about 30. To address this limitation, recent studies have explored fine-tuning for image generation under limited data (Ruiz et al., 2023; Moon et al., 2022; Zhu et al., 2022; Hur et al., 2024;

---

*Corresponding author.

Yang et al., 2024; Lu et al., 2023; Zhang et al., 2025). For example, Ruiz et al. (2023) applied fine-tuning to transfer knowledge from models pre-trained on large-scale external datasets, which allows the model to synthesize high-quality images using only a few target examples. However, these approaches heavily rely on the similarity between the source (i.e., large-scale external datasets) and the target dataset (i.e., the limited dataset) (Hur et al., 2024). This reliance hinders the broader adoption of generative diffusion models in data-sensitive fields such as medicine (Kazerouni et al., 2022). More critically, Moon et al. (2022) observed that when limited data are used to fine-tune a pretrained diffusion backbone, the model suffers from *overfitting*, which means that it memorizes individual training examples rather than captures the underlying data distribution, and this results in near-duplicate outputs and reduced diversity (Webster et al., 2019; Karras et al., 2020). This problem is particularly severe under limited data settings, where the scarcity of training samples makes the model prone to memorization rather than generalization.

To further illustrate this overfitting phenomenon, we conduct an empirical study to examine how training data size influences their convergence behavior. Specifically, we investigate the performance dynamics by training a denoising diffusion probabilistic model (DDPM) (Ho et al., 2020) on subsets of FFHQ ($64 \times 64$) (Karras et al., 2018). We measure the quality by computing Fréchet inception distance (FID) (Heusel et al., 2017) between 50k generated images and all available training images. As shown in Fig. 1a, the FID curve exhibits a "U-shaped" trend: it decreases in the early stages, reaches a minimum FID, and then worsens as training continues; smaller datasets yield an earlier turning point and a higher final FID, which clearly indicates overfitting. It is worth noting that previous work (Karras et al., 2020) reported similar convergence behavior for GANs. Furthermore, we also evaluated DDPM on the CelebA-HQ ($64 \times 64$) (Liu et al., 2015) dataset. The results, shown in Fig. 1b, are consistent with the above findings that the FID curves also exhibit a U-shaped trend. For completeness, we also illustrate that the training loss decreases monotonically in all cases (as shown in Fig. 1c), while at the same time the FID curve exhibits a U-shaped pattern, which indicates that overfitting indeed exists.

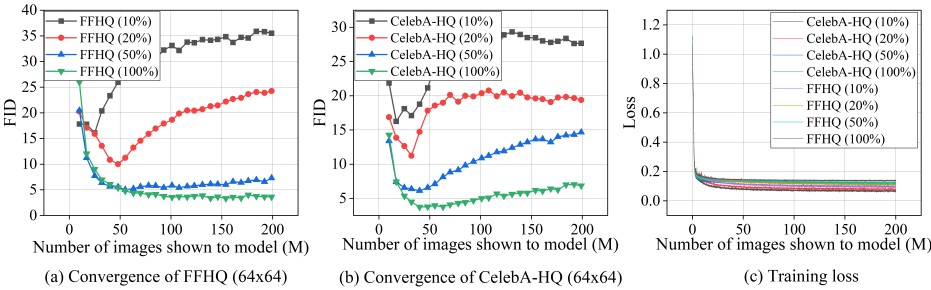

(a) Convergence of FFHQ (64x64)          (b) Convergence of CelebA-HQ (64x64)          (c) Training loss

Figure 1: Evidence of overfitting in diffusion models with limited data. (a, b) FID curves of DDPM on FFHQ (64×64) and CelebA-HQ (64×64) datasets, both exhibiting a "U-shaped" trend where smaller datasets yield earlier turning points and higher final FID. Percentages (e.g., $50\%$) indicate the fraction of training data used. (c) Training loss decreases monotonically across all cases.

For classification models, a wide range of methods have been developed to address the problem of overfitting. These approaches can be broadly divided into two categories: (1) regularization-based techniques, such as $L_1/L_2$ penalties (Tibshirani, 1996; Ng, 2004); and (2) data augmentation strategies, such as Cutout (DeVries & Taylor, 2017), Mixup (Zhang et al., 2018), and CutMix (Yun et al., 2019). However, most of these methods are tailored to classification objectives and cannot be directly transferred to handle diffusion models due to the following two main reasons. **(R1)** Regularization-based techniques are primarily designed to improve the generalization of *decision boundaries* in classification models; they provide limited benefit when the goal is to capture the underlying data distribution, as in diffusion models. **(R2)** Augmentation-based techniques are typically static and rule-driven. These methods can not adaptively constrain distributional shift, and may even exacerbate the discrepancy by pushing the training marginal distribution further away from the true data distribution. As a result, the model could learn off-distribution artifacts and reproduce them at generation time, where this problem is called "augmentation leakage" in (Karras et al., 2020).

In this paper, our proposed method is inspired by *Wasserstein Distributionally Robust Optimization* (WDRO) (Gao & Kleywegt, 2023; Sinha et al., 2018; Huang & Ding, 2025), an elegant and pow-

erful mathematical framework from the field of robust optimization. A major advantage is that it operates directly on data distribution and adaptively expands the support of the training distribution while remaining close to the true data distribution. Specifically, WDRO replaces *empirical risk minimization* (ERM) on the limited data data distribution $p_{\text{data}}$ with optimization against the *worst-case* distribution in a *Wasserstein uncertainty set*

$$\mathcal{U}_\rho(p_{\text{data}}) = \{p : \mathcal{W}_{\mathbf{c}}(p, p_{\text{data}}) \leq \rho\}, \tag{1}$$

a $\rho$-neighborhood of the distribution $p_{\text{data}}$ under the Wasserstein metric $\mathcal{W}_{\mathbf{c}}(\cdot, \cdot)$ (see Section 2.2 for a formal definition). WDRO has been proven to effectively mitigate overfitting in supervised learning (e.g., adversarial training (Liu et al., 2025) and continual learning (Wang et al., 2023c)), by dynamically adjusting the data distribution. Conceptually, WDRO can be viewed as an adaptive method for *support expansion*: rather than fitting only the narrow support of $p_{\text{data}}$ (a key source of overfitting in limited data settings), the learner is trained to perform well over a neighborhood of distributions within a transportation budget $\rho$. Therefore, under the WDRO perspective, a natural question arises:

*Can the idea of "adaptive support expansion" in WDRO be applied to diffusion models to enlarge the effective training support, with the goal of improving generative quality while mitigating overfitting in limited data settings?*

## 1.1 OUR MAIN CONTRIBUTIONS

To address the above question, we propose a "**WDRO Inspired training method for Diffusion model under Limited Data (WILD-Diffusion)**", a **plug-and-play** training framework that leverages WDRO to dynamically expand the support of the limited data distribution, which can mitigate overfitting and enhance generative performance. It is worth noting that the idea of DRO has recently been introduced into diffusion models (Wang et al., 2025a); however, this work addresses a different problem about diffusion models, which focuses on the training and sampling distribution mismatch issue rather than limited data generation. Specifically, we apply WDRO to the diffusion problem, where the objective can be formulated as

$$\underset{\theta}{\text{minimize}} \sup_{p \in \mathcal{U}_\rho(p_{\text{data}})} \mathbb{E}_p[\ell(\theta; \mathbf{x}, t)], \tag{2}$$

where the uncertainty set $\mathcal{U}_\rho(p_{\text{data}})$ is defined in Eq. (1), $\theta$ denotes the model parameters, $(\mathbf{x}, t)$ are the diffusion training inputs (data $\mathbf{x}$ and time point $t$), and $\ell(\theta; \mathbf{x}, t)$ represents the diffusion training loss function, which will be formally defined in a later section (see Eq. (6)). The solution of the problem (2) guarantees reliable performance against data distributions that are distance $\rho$ away

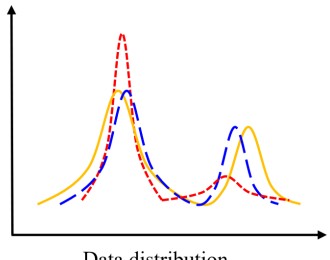

Figure 2: Illustration of support expansion in a 1D setting. Yellow (solid): true distribution; Red (dotted): limited data distribution; Blue (dashed): distribution induced by **WILD-Diffusion**, which expands the support of the limited data distribution toward the true distribution and narrows the gap.

from the limited data distribution $p_{\text{data}}$. Roughly speaking, the solution of problem (2) is expected to expand the support toward the underlying data distribution and narrow the gap (as illustrated in Figure 2), which in turn mitigates overfitting and improves sample quality under limited data settings.

Nevertheless, efficiently implementing this idea within diffusion training is not straightforward, as it involves two major challenges. **(C1)** Because both diffusion training and the computation of the Wasserstein distance are computationally expensive, the first difficulty is to ensure the inner maximization tractable while preserving overall training efficiency. **(C2)** Since WDRO is inherently a min–max optimization problem with notoriously difficult convergence, another critical challenge is to establish theoretical convergence guarantee for the WILD-Diffusion framework. To tackle these challenges, we build on the surrogate loss idea (Blanchet & Murthy, 2019) and reformulate problem (2) as an approximate optimization problem that is tractable in Euclidean space. This reformulation ensures the otherwise intractable inner maximization can be efficient computation, and we further propose a "Bi-level Interval Update" strategy to derive a practical approximate solution. Specifically, the strategy alternates between parameter updates on the current mixed training set (i.e., original samples and their adversarial counterparts) and distribution interval updates through

worst-case sample generation. Furthermore, we establish convergence guarantee for the proposed WILD-Diffusion method. Unlike prior work (Lee et al., 2022), which analyzes the convergence of standard diffusion models, the incorporation of WDRO requires an additional technical step: we prove an upper bound for the worst-case objective (Lemma 3.5), which is essential for achieving the convergence of our proposed WILD-Diffusion.

The experiments on a variety of diffusion architectures (DDPM++, NCSN++, and ADM) and datasets (CIFAR-10, LSUN-Church, CelebA-HQ, and FFHQ) suggest the effectiveness of our method. With WILD-Diffusion, we can achieve more than a $10\%$ reduction in FID using only $20\%$ of the training data across all datasets. In addition, our method achieves state-of-the-art FID with as few as $100$ images, in both pretrained and non-pretrained settings.

## 2 BACKGROUND

In this section, we first review the background of diffusion-based generative models, outlining their key formulations and training objectives. We then introduce the concept of Wasserstein distance, which plays a central role in the formulation of our WILD-Diffusion framework. Due to space limitations, additional related work is provided in the Appendix A.

### 2.1 DIFFUSION-BASED GENERATIVE MODELS

Suppose we are given a dataset $\{\mathbf{x}_i\}_{i=1}^n$, where each data point is independently drawn from an underlying data distribution with positive density $p_{\text{data}}(\mathbf{x})$. We slightly abuse notation by using a measure and its density interchangeably when the context is clear. The forward process is to construct a process $\{\mathbf{x}(t)\}_{t=0}^T$ indexed by a continuous time variable $t \in [0, T]$. Note that the process starts from $\mathbf{x}(0) \sim p_{\text{data}}(\mathbf{x})$ and evolves to $\mathbf{x}(T) \sim p_T(\mathbf{x})$, where $p_T$ typically denotes a simple prior distribution, such as a standard Gaussian (Ho et al., 2020). According to (Song et al., 2021b), the forward diffusion process can be modeled as a stochastic differential equation (SDE):

$$\mathrm{d}\mathbf{x} = \mathbf{f}(\mathbf{x}, t) \, \mathrm{d}t + g(t) \, \mathrm{d}\mathbf{w}, \tag{3}$$

where $\mathbf{f}(\cdot, t) : \mathbb{R}^d \to \mathbb{R}^d$ is called the *drift* coefficient of $\mathbf{x}(t)$, $g(\cdot) : \mathbb{R} \to \mathbb{R}$ is a scalar function known as the diffusion coefficient of $\mathbf{x}(t)$, $\mathbf{w}$ is the standard Wiener process (a.k.a., Brownian motion), and $\mathrm{d}t$ represents a negative infinitesimal timestep. Importantly, for any forward diffusion process in the form of Eq.(3), Anderson (1982) showed that it could be reversed by solving the following reverse-time SDE:

$$\mathrm{d}\mathbf{x} = \left[\mathbf{f}(\mathbf{x}, t) - g(t)^2 \nabla_{\mathbf{x}} \log p_t(\mathbf{x})\right] \mathrm{d}t + g(t) \, \mathrm{d}\bar{\mathbf{w}}, \tag{4}$$

where $\bar{\mathbf{w}}$ is a standard Wiener process when time flows backwards, and the gradient of the log probability density with respect to the data, $\nabla_{\mathbf{x}} \log p_t(\mathbf{x})$, is the *(Stein) score* (Liu et al., 2016). Moreover, Song et al. (2021b) proved the existence of an ordinary differential equation (ODE), namely the *probability flow ODE*, whose trajectories have the same marginals as the reverse-time SDE (4). The probability flow ODE is expressed as:

$$\mathrm{d}\mathbf{x} = \left[\mathbf{f}(\mathbf{x}, t) - \frac{1}{2}g(t)^2 \nabla_{\mathbf{x}} \log p_t(\mathbf{x})\right] \mathrm{d}t. \tag{5}$$

Note that if the score of the marginal distributions, $\nabla_{\mathbf{x}} \log p_t(\mathbf{x})$, is known for all $t \in [0, T]$, then the reverse diffusion process can be derived from Eq. (5) and subsequently simulated to generate samples from $p_{\text{data}}(\mathbf{x})$. Specifically, a time-dependent score model $\mathbf{s}_\theta(\mathbf{x}, t)$ is trained to estimate the score function, which yields the following training objective:

$$\ell(\theta, \mathbf{x}, t) = \lambda(t) \cdot \|\mathbf{s}_\theta(\mathbf{x}, t) - \nabla_{\mathbf{x}} \log p_t(\mathbf{x})\|_2^2, \tag{6}$$

where $\lambda(t) : [0, T] \to \mathbb{R}_+$ is a positive weighting function (Yang et al., 2023b).

### 2.2 WASSERSTEIN DISTANCE

The *Wasserstein distance*, which originates from the theory of *optimal transport* (Peyré et al., 2019; Villani et al., 2008), has been widely adopted in machine learning (Sinha et al., 2018; Kolouri et al., 2017). Let $\mathcal{X} \subset \mathbb{R}^d$ denote the sample space. For $\mathbf{x}, \mathbf{x}' \in \mathcal{X}$, the transportation cost $\mathbf{c}$ associated with moving mass from $\mathbf{x}$ to $\mathbf{x}'$ is defined as (Volpi et al., 2018)

$$\mathbf{c}(\mathbf{x}, \mathbf{x}') := \frac{1}{2}\|\mathbf{x} - \mathbf{x}'\|_2^2. \tag{7}$$

As the $L_2$ norm is the standard choice in optimal transport, we confine our analysis to this setting. Given two probability measures $P$ and $Q$ supported on $\mathcal{X}$, let $\Pi(P, Q)$ denote the set of couplings between $P$ and $Q$, i.e., measures $M$ on $\mathcal{X} \times \mathcal{X}$ with marginals $P$ and $Q$. Then, the *Wasserstein distance* between $P$ and $Q$ is defined as

$$\mathcal{W}_{\mathbf{c}}(P, Q) := \inf_{M \in \Pi(P, Q)} \mathbb{E}_M\left[\mathbf{c}(\mathbf{x}, \mathbf{x}')\right]. \tag{8}$$

## 3 Method

In this section, we present WILD-Diffusion, a WDRO inspired framework designed to enable effective training of diffusion models in limited data settings. A highlight of WILD-Diffusion is that it dynamically leverages WDRO to construct *worst-case distributions* that lie close to the limited data distribution (in Wasserstein distance), which expands the support of the training distribution and improves sample diversity, and consequently relieves the negative impact of overfitting. Moreover, our framework is flexible and can be combined with a wide range of baseline methods. We first present our WILD-Diffusion framework in Section 3.1. Next, we provide the convergence analysis of our proposed approach in Section 3.2.

### 3.1 WILD-Diffusion Framework

Wasserstein Distributionally Robust Optimization (WDRO) (Kuhn et al., 2019; Rahimian & Mehrotra, 2019b; Sinha et al., 2018) formulates robust decision-making under uncertainty by optimizing for the worst-case over all probability distributions within a Wasserstein ball. The Wasserstein ball consists of all distributions whose distance from the limited data distribution does not exceed a given threshold (recall $\rho$ in Eq. (2)). In our WILD-Diffusion framework, we assume that the true data distribution lies in a Wasserstein uncertainty set (1), i.e., $\mathcal{U}_\rho(p_{\text{data}}) = \{p : \mathcal{W}_{\mathbf{c}}(p, p_{\text{data}}) \le \rho\}$. This formulation captures the distributional uncertainty arising from limited data, which is particularly severe when the sample size is small because the limited data distribution poorly approximates the true underlying distribution (see Fig. 2). Recall the optimization objective (2), the inner `sup` over the Wasserstein uncertainty set enforces the model to cope with increasingly harder perturbations of the limited data distribution. Namely, this strategy can guide the model to learn some new samples and therefore prevents memorization and thus mitigates overfitting.

In general, the worst-case optimization that involves the `sup` operator within the Wasserstein ball is computationally challenging for two main reasons: (i) the Wasserstein ball encompasses a rich family of probability distributions, making the inner maximization problem inherently infinite-dimensional; and (ii) computing the Wasserstein distance itself is computationally expensive even in approximate forms. While these challenges already arise for relatively simple models, they become particularly severe in the context of diffusion models. To handle the inner maximization problem in (2), we adopt the strong duality property given in (Gao & Kleywegt, 2023, Theorem 1) and obtain its dual formulation. Suppose $\mathcal{X} \subset \mathbb{R}^d$ is the sample space. Given a fixed penalty parameter $\gamma \ge 0$, the worst-case loss in Eq. (2) can be reformulated as

$$\underset{\theta}{\text{minimize}} \left\{ \mathcal{L}(\theta) := \sup_p \left\{ \mathbb{E}_p[\ell(\theta; \mathbf{x}, t)] - \gamma \mathcal{W}_{\mathbf{c}}(p, p_{\text{data}}) \right\} = \mathbb{E}_{p_{\text{data}}}\left[ \phi_\gamma(\theta; \mathbf{x}, t) \right] \right\}, \tag{9a}$$

$$\text{where} \quad \phi_\gamma(\theta; \mathbf{x}, t) := \sup_{\mathbf{x}' \in \mathcal{X}} \left\{ \ell(\theta; \mathbf{x}', t) - \gamma \mathbf{c}(\mathbf{x}', \mathbf{x}) \right\}, \tag{9b}$$

is the surrogate loss (Blanchet & Murthy, 2019; Volpi et al., 2018) that replace the usual diffusion loss $\ell(\theta; \mathbf{x}, t)$ (i.e., Eq. (6)). Here, the penalty parameter $\gamma$ controls the degree of support expansion; it balances fidelity to the training data and robustness to distributional shifts. Since $p_{\text{data}}$ is unknown, the penalty problem (9a) is solved by replacing $p_{\text{data}}$ with the empirical distribution $\hat{p}_n$, where $n$ is the sample size.

**Remark 3.1** *Eq. (9a) gives the dual formulation of Eq. (2), i.e., both problems share the same optimal value. The advantage of this reformulation is that we can ignore the complicated uncertainty set $\mathcal{U}_\rho(p_{data})$. Instead, we only add a surrogate loss $\phi_\gamma(\theta; \mathbf{x}, t)$ to the Eq. (9a), which yields a more succinct formulation for optimizing the problem. However, the solution to Eq. (9a) is non-trivial; we provide further details on its optimization in the following discussion.*

In order to solve the duality formulation (9a), we can now perform stochastic gradient descent on the surrogate loss $\phi_\gamma$. Specifically, suppose that the loss $\ell(\theta; \mathbf{x}, t)$ satisfies the Lipschitz smoothness conditions (Boyd & Vandenberghe, 2004) and that the surrogate loss is strongly concave. Under these conditions, we have

$$\nabla_\theta \phi_\gamma(\theta; \mathbf{x}, t) = \nabla_\theta \ell(\theta; (\mathbf{x}^*, t)) \quad \text{where} \quad \mathbf{x}^* = \underset{\mathbf{x}' \in \mathcal{X}}{\operatorname{argmax}} \left\{ \ell(\theta; \mathbf{x}', t) - \gamma \mathbf{c}(\mathbf{x}', \mathbf{x}) \right\}. \tag{10}$$

Computing the gradient of the surrogate loss $\phi_\gamma$ for a given sample $\mathbf{x}$ requires solving the inner maximization problem to obtain $\mathbf{x}^*$. Notably, we observe that $\mathbf{x}^*$ is similar to an adversarial perturbation of $\mathbf{x}$ under the current model $\theta$. Following the intuition of *adversarial training* (Madry et al., 2018), we propose a "Bi-level Interval Update" strategy for WILD-Diffusion. The difference from adversarial training is that, while adversarial training typically generates adversarial examples within a fixed norm ball, our approach imposes a soft constraint via the penalty parameter $\gamma$, which governs distributional robustness at the support level. The strategy couples two updates. **(I) Parameter update level**. The model parameters $\theta$ are updated at every training iteration using the current training set. **(II) Distribution (sample) update level**. Every $m$ epochs we refresh the WDRO-induced "worst-case" samples via gradient ascent and mix them with the real data to form the augmented training distribution. Between distribution updates, the worst-case samples are kept fixed. Specifically, at the sample update level, for each training example we first draw an initial point $\mathbf{x}_i^0$ from the data distribution $p_{\text{data}}$. We then iteratively update it through the injection of adversarial perturbations, which produces an adversarial variant as defined by the following update rule:

$$\mathbf{x}_i^k \leftarrow \mathbf{x}_i^{k-1} + \zeta \nabla_{\mathbf{x}} \{ \ell(\theta; \mathbf{x}_i^{k-1}, t) - \gamma \mathbf{c}(\mathbf{x}_i^{k-1}, \mathbf{x}_i^0) \}, \tag{11}$$

where $\zeta$ denotes the step size and $k = 1, \ldots, K$ indexes the iterations. At the parameter update level, the model parameters $\theta$ are updated at every training step by performing stochastic gradient descent on the loss $\ell(\theta; \mathbf{x}, t)$, where the training sets is a mixture of the original samples and their adversarial counterparts. Algorithm 1 presents the proposed WILD-Diffusion algorithm, which offers the flexibility to incorporate a variety of baseline methods, since it operates on the data distribution without requiring changes to the model architectures. In addition, we take $S_{\text{w}}$ epochs to train the model on the limited dataset as a warmup stage. The warmup stage yields a stable initialization before incorporating worst-case samples. Starting from a well-initialized state enables the model to produce more informative gradients used in Eq. (11). In practice, We allocate 20% of the total training epochs to the warmup stage.

## 3.2 CONVERGENCE ANALYSIS

In this section, we establish the convergence guarantee for the proposed **WILD-Diffusion** method. In contrast to prior work (Lee et al., 2022), which focuses on standard diffusion models, our analysis must account for the additional complexity introduced by WDRO. To this end, we establish a upper bound for the worst-case objective (Lemma 3.5), which enables the convergence proof of WILD-Diffusion. We first make the following assumptions (i.e., Assumption 3.2 and 3.3) on the probability density $p_{\text{data}}$ and the score estimate $\mathbf{s}_\theta(\mathbf{x}, t)$ (defined in Section 2), which will be used throughout the analysis.

**Assumption 3.2** *Assume that $p_{data}$ satisfies the log-Sobolev inequality with constant $C_{\text{lS}} > 1$; $\log p_{data}$ is $L$-Lipschitz for some $L \geq 1$; $p_{data}$ has finite first and second moments.*

**Assumption 3.3** *Suppose that $\mathbf{s}_\theta(\mathbf{x}, t)$ is $L_{\text{s}}$-Lipschitz in its first argument with $L_{\text{s}} \geq 1$, and the error in score estimate $\ell(\theta; \mathbf{x}, t)$ is uniformly bounded by a given parameter $\varepsilon > 0$.*

**Remark 3.4** *Assumptions 3.2 and 3.3, also adopted in Lee et al. (2022), are standard assumptions in analyses of score-based diffusion models. In particular, the Lipschitz assumption on $p_{data}$ is used to ensure the existence of a unique strong solution to the reverse-time SDE (Eq. 4) (Block et al., 2020; Øksendal, 2003). The detailed definition of the log-Sobolev inequality is given in Appendix D. Building on the above assumptions, we derive an upper bound for the optimization objective in Eq. (2), as stated in Lemma 3.5, which is a essential condition for the convergence analysis of WILD-Diffusion.*

---

**Algorithm 1** WILD-Diffusion

---

**Input:** Training datasets $\{\mathbf{x}_i\}_{i=1}^n$; Initialized model parameter $\theta_0$, learning rate $\eta$, step size $\zeta$, number of iterations $K$ in inner optimization, interval parameter $m$, total diffusion steps $T$, the number of epochs $S$, and the number of warmup epochs $S_{\mathrm{w}}$.

**Output:** Final diffusion model parameter $\theta$.

1: $\theta \leftarrow \theta_0$      */* Initialize model */*
2: **for** $s = 1, \ldots, S_{\mathrm{w}}$ **do**
3:      */* Take $S_w$ epochs to train the model as the warmup */*
4:      **for** $i = 1, \ldots, n$ **do**
5:          Sample $t \sim \mathrm{Uniform}(\{1, \ldots, T\})$
6:          $\theta \leftarrow \theta - \eta \nabla_\theta \ell(\theta; \mathbf{x}_i, t)$
7:      **end for**
8: **end for**
9: **for** $s = S_{\mathrm{w}} + 1, \ldots, S$ **do**
10:      **if** $(s \bmod m)$==0 **then**
11:          */* Support Expansion via WDRO */*
12:          $\mathcal{D} \leftarrow \{\}$
13:          **for** $i = 1, \ldots, n$ **do**
14:              $\mathbf{x}_i^0 \leftarrow \mathbf{x}_i, t \sim \mathrm{Uniform}(\{1, \ldots, T\})$
15:              **for** $k = 1, \ldots, K$ **do**
16:                  $\mathbf{x}_i^k \leftarrow \mathbf{x}_i^{k-1} + \zeta \nabla_{\mathbf{x}}\{\ell(\theta; \mathbf{x}_i^{k-1}, t) - \gamma \mathbf{c}(\mathbf{x}_i^{k-1}, \mathbf{x}_i^0)\}$      */* Distribution (sample) update via Eq. (11) */*
17:              **end for**
18:              $\mathcal{D} \leftarrow \mathcal{D} \cup \{\mathbf{x}_i^K\}$      */* Save worst-case samples */*
19:          **end for**
20:      **end if**
21:      **for** $i = 1, \ldots, n$ **do**
22:          Sample $\mathbf{x}_i' \sim \mathcal{D}, t \sim \mathrm{Uniform}(\{1, \ldots, T\})$
23:          $\theta \leftarrow \theta - \eta \nabla_\theta \{\ell(\theta; \mathbf{x}_i, t) + \ell(\theta; \mathbf{x}_i', t)\}$      */* Parameter update */*
24:      **end for**
25: **end for**

---

**Lemma 3.5** *Under Assumption 3.3, for any fixed $\tau > 0$, the following inequality holds with probability at least $1 - e^{-\tau}$, uniformly over all $\rho \geq 0$ and $\gamma \geq 0$*

$$\sup_{p:\mathcal{W}_{\mathbf{c}}(p, p_{data}) \leq \rho} \mathbb{E}_p[\ell(\theta; \mathbf{x}, t)] \leq \gamma\rho + \mathbb{E}_{\hat{p}_n}[\phi_\gamma(\theta; \mathbf{x}, t)] + O\left(\sqrt{\tfrac{\tau}{n}}\right). \tag{12}$$

Here, $n$ is the sample size, $\tau$ is the confidence parameter, and $\hat{p}_n$ is the empirical distribution of the samples from $p_{\mathrm{data}}$. We adopt the *total variation distance* $\mathrm{D}_{\mathrm{TV}}(\cdot, \cdot)$ to quantify convergence. Given two distributions $p$ and $q$, the total variation distance is defined as $\mathrm{D}_{\mathrm{TV}}(p, q) = \frac{1}{2}\int |p(\mathbf{x}) - q(\mathbf{x})|\mathrm{d}\mathbf{x}$, which measures the maximum discrepancy between two distributions. Before presenting the convergence result of WILD-Diffusion, we first provide an outline of the proof. Namely, let $q_t$ denote the reverse process with the estimated score. We define the *bad set* $B_t$ as $B_t = \left\{ \mathbf{x} \mid \sup_{p:\mathcal{W}_{\mathbf{c}}(p, p_{\mathrm{data}}) \leq \rho} \mathbb{E}_p\big[\|\mathbf{s}_\theta(\mathbf{x}, t) - \nabla_{\mathbf{x}} \log p_t(\mathbf{x})\|^2\big] > \varepsilon_B \right\}$ for some $\varepsilon_B$ to be chosen, and define $\bar{q}_t$ as the reverse process with the estimated score except in $B_t$. Hence, the convergence proof can be divided into two parts by applying the triangle inequality

$$\mathrm{D}_{\mathrm{TV}}(q_t, p_t) \leq \mathrm{D}_{\mathrm{TV}}(\bar{q}_t, p_t) + \mathrm{D}_{\mathrm{TV}}(q_t, \bar{q}_t). \tag{13}$$

Since (Lee et al., 2022) established the bound $\mathrm{D}_{\mathrm{TV}}(\bar{q}_t, p_t) \leq \varepsilon_\chi^2 < 1$, with $\varepsilon_\chi^2$ denoting the corresponding error term, the main task is therefore to control the second term $\mathrm{D}_{\mathrm{TV}}(q_t, \bar{q}_t)$. This is established in Theorem 3.6.

**Theorem 3.6** (*Convergence of WILD-Diffusion*). *Suppose Assumptions 3.2 and 3.3 hold, and Lemma 3.5 applies. If we run the SDE (Eq. 4) starting from a Gaussian distribution for time $T = \Theta\left(\max\left\{\log(C_{\mathrm{lS}}d), C_{\mathrm{lS}}\log\left(\frac{2}{\varepsilon_\chi^2}\right)\right\}\right)$ with step size $h = \Theta\left(\frac{\varepsilon_\chi^2}{C_{\mathrm{lS}}(C_{\mathrm{lS}}+d)\max\{L^2, L_s^2\}}\right)$, then*

*the final sampling distribution $q_0$ satisfies*

$$\mathrm{D_{TV}}(q_0, \bar{q}_0) \leq O\left(\sqrt{\gamma\rho + \mathbb{E}_{\hat{p}_n}[\phi_\gamma(\theta; \mathbf{x}, t)] + O\left(\sqrt{\frac{1}{n}}\right)} \cdot \frac{C_{\mathrm{lS}}^{5/2}(C_{\mathrm{lS}}+d)(L^2+L_s^2)\left(1+\log\left(\frac{2}{\varepsilon_\chi^2}\right)\right)}{\varepsilon_\chi^3}\right). \quad (14)$$

*For simplicity, we denote the upper bound in Eq. (14) by $\mathrm{D}_{ub}$. Thus, $\mathrm{D_{TV}}(q_0, p_{data}) \leq \varepsilon_\chi^2 + \mathrm{D}_{ub}$.*

The complete proof is provided in Appendix B.3. Theorem 3.6 establishes a convergence guarantee for WILD-Diffusion under standard assumptions. Specifically, the total variation distance between the generated distribution $q_0$ and the limited data distribution $p_{\mathrm{data}}$ is bounded by the sum of two terms: an estimation error term $\varepsilon_\chi^2$, which arises from the approximation of the score function, and a sampling error term $\mathrm{D_{TV}}(q_0, \bar{q}_0)$, which is due to the numerical computation of the reverse SDE. Notably, when the robustness budget $\rho \to 0$ and the sample size $n \to \infty$, the bound recovers the result of (Lee et al., 2022), which showed that

$$\mathrm{D_{TV}}(q_0, p_{\mathrm{data}}) \leq \varepsilon_\chi^2 + O\left(\sqrt{\varepsilon} \cdot C_{\mathrm{lS}}^{5/2}(C_{\mathrm{lS}}+d)(L^2+L_s^2)\left(1+\log\left(\frac{2}{\varepsilon_\chi^2}\right)\right)\varepsilon_\chi^{-3}\right).$$

This suggests that our convergence guarantee can be regarded as a generalization of the result in (Lee et al., 2022) to the more complicated distributionally robust setting (see Appendix B.4 for details).

# 4 EXPERIMENTS

In this section, we first present a hyper-parameter sensitivity analysis to investigate the key factors influencing the performance of our method, as detailed in Section 4.1. Next, we compare our approach with state-of-the-art diffusion model baselines on widely-used benchmark datasets in Section 4.2. In Section 4.3, we further demonstrate that our method performs well on few-shot datasets. It is worth noting that in generative modeling, the few-shot setting differs from the limited data regime: the former typically involves adapting a pretrained model to a new distribution with only a handful of samples (tens to hundreds), whereas the latter refers to training on a small dataset of only thousands of samples without access to large-scale pretraining (Abdollahzadeh et al., 2023). Finally, we conduct the ablation studies in Section 4.4.

**Experimental Setting.** In line with previous works (Wang et al., 2023a; Zhao et al., 2020; Karras et al., 2020), we conduct experiments on standard benchmarks, where subsets of the training data are randomly selected. For the *limited data* setting, we adopt CIFAR-10 ($32 \times 32$) (Krizhevsky et al., 2009), FFHQ ($64 \times 64$) (Karras et al., 2019), CelebA-HQ ($64 \times 64$) (Karras et al., 2018), and LSUN-Church ($256 \times 256$) (Yu et al., 2015). For the *few-shot* setting, we adopt the 100-shot datasets ($256 \times 256$)—Obama, Grumpy Cat, and Panda (Zhao et al., 2020)—and AnimalFace ($256 \times 256$; cats and dogs) (Si & Zhu, 2011). We implement our method on the current start-of-the-art diffusion framework EDM (Karras et al., 2022), which integrates DDPM++ (Song et al., 2021b), NCSN++ (Song et al., 2021b), and ADM (Dhariwal & Nichol, 2021). DDPM++ is our default backbone model for training low-resolution (i.e., 32×32 and 64×64) datasets, while ADM coupling with Stable Diffusion (Rombach et al., 2022) is our backbone model for training high-resolution (i.e., 256×256) datasets. We evaluate image generation quality using Fréchet Inception Distance (FID) (Heusel et al., 2017). Following Karras et al. (2022; 2020), FID is computed between 50k generated samples and the full set of training images. The detailed experimental settings are provided in Appendix C.

## 4.1 SENSITIVITY OF HYPER-PARAMETER

In this section, we investigate the sensitivity of our method to key hyper-parameters. In particular, the interval parameter $m$ (in Algorithm 1) plays a crucial role, as it can substantially influence both generation quality and training efficiency. To assess its effect, we conduct a series of experiments by varying $m$ over $\{5, 10, 20, 30, 40, 50, 100\}$ on the FFHQ dataset with 50% training data. Figure 3 shows that increasing $m$ reduces the total training time while degrading generative performance (higher FID). This reveals a clear trade-off between efficiency and quality. Taking both training efficiency and generative quality into account, we set $m = 20$ as the default choice in all experiments.

In addition, to better interpret the influence of injected adversarial perturbations (see Eq. 11), we examine our method on the FFHQ dataset with 50% of the training data across the number of steps,

step size, and penalty strength. When studying one factor, the others are fixed at their best values. From Figure 4, several observations can be drawn: (1) Increasing the number of steps $K$ improves performance up to $K = 5$, after which the gains diminish (Figure 4a); (2) The step size $\eta = 0.01$ achieves the best balance, while both smaller and larger values degrade performance (Figure 4b); (3) The penalty parameter $\gamma$ is relatively stable, with $\gamma = 1$ performing best (Figure 4c). In summary, we adopt these configurations as the default in all experiments.

## 4.2 EXPERIMENTS ON LIMITED DATA GENERATION

In this section, we compare our method with state-of-the-art diffusion approaches on both low-resolution (Table 1) and high-resolution (Table 5) benchmarks. For the low-resolution setting, we evaluate on CIFAR-10, CelebA, CelebA-HQ, and FFHQ. Specifically, the baselines include EDM-DDPM++ (Karras et al., 2022), EDM-NCSN++ (Karras et al., 2022), EDM-ADM (Karras et al., 2022), (Wang et al., 2025a), Patch Diffusion (Wang et al., 2023a), and DeepCache (Ma et al., 2024). We include Patch Diffusion and DeepCache to illustrate that WILD-Diffusion serves as a plug-and-play framework that is "orthogonal" to these methods; moreover, they can be seamlessly combined to achieve more promising performance in practice. We incorporate our WILD-Diffusion into these diffusion methods to assess its performance across datasets. For completeness, we also compare with data-efficient GAN-based approaches, including BigGAN (Brock et al., 2019), StyleGAN-v2 (Karras et al., 2019), DiffAugment (Zhao et al., 2020), and CR-BigGAN (Zhang et al., 2020).

Table 1: FID results on low-resolution datasets. FID (lower is better) is computed with $50k$ samples. The numerical results of the baseline methods are taken from the original papers. "-" indicates that the result is not reported in the original paper (Zhao et al., 2020). The notation "(-$\Delta$%)" indicates percentage decreases compared to the baseline. "$\Delta$% data" refers to randomly selecting "$\Delta$%" of the training data from the dataset, and "*cond.*" denotes the class-conditional setting. The best-performing results are highlighted in **bold**.

| Dataset | Method | 20% data | 50% data | 100% data |
|---|---|---|---|---|
| | BigGAN (Brock et al., 2019) | 21.58 | - | 9.59 |
| | StyleGAN-v2 (Karras et al., 2019) | 23.08 | - | 11.07 |
| | CR-BigGAN (Zhang et al., 2020) | 20.62 | - | 9.06 |
| | BigGAN+DiffAugment (Zhao et al., 2020) | 14.04 | - | 8.70 |
| | EDM-DDPM++ (Karras et al., 2022) | 13.91 | 6.62 | 1.97 |
| | (Wang et al., 2025a) | 13.63 | 6.49 | - |
| | + WILD-Diffusion | 12.14 (-12.72%) | 6.02 (-9.08%) | 1.93 (-2.03%) |
| | EDM-DDPM++ (*cond.*) (Karras et al., 2022) | 12.33 | 6.03 | 1.79 |
| CIFAR-10 | + WILD-Diffusion (*cond.*) | **10.89** (-11.68%) | **5.37** (-10.95%) | **1.71** (-4.47%) |
| ($32 \times 32$) | EDM-NCSN++ (Karras et al., 2022) | 13.68 | 6.53 | 2.02 |
| | + WILD-Diffusion | 12.08 (-11.70%) | 5.97 (-8.58%) | 1.98 (-1.98%) |
| | Patch Diffusion (Wang et al., 2023a) | 12.53 | 6.42 | 2.47 |
| | + WILD-Diffusion | 11.78 (-5.99%) | 6.07 (-5.45%) | 2.38(-3.64%) |
| | DeepCache (Ma et al., 2024) | 15.33 | 9.31 | 4.35 |
| | + WILD-Diffusion | 13.96 (-8.94%) | 8.72 (-6.34%) | 4.21 (-3.37%) |
| FFHQ | EDM-DDPM++ (Karras et al., 2022) | 10.02 | 5.21 | 2.60 |
| | + WILD-Diffusion | 8.57 (-14.47%) | 4.68 (-10.17%) | 2.53 (-2.70%) |
| ($64 \times 64$) | EDM-NCSN++ (Karras et al., 2022) | 9.38 | 5.04 | 2.57 |
| | + WILD-Diffusion | **7.89** (-15.88%) | **4.60** (-8.73%) | **2.54** (-1.16%) |
| CelebA-HQ | EDM-DDPM++ (Karras et al., 2022) | 11.86 | 6.11 | 3.73 |
| | + WILD-Diffusion | 10.22 (-13.83%) | 5.55 (-9.17%) | **3.63** (-2.68%) |
| ($64 \times 64$) | EDM-NCSN++ (Karras et al., 2022) | 11.63 | 5.81 | 3.70 |
| | + WILD-Diffusion | **10.07** (-13.41%) | **5.36** (-7.75%) | 3.65 (-1.35%) |

The results for the low-resolution benchmarks are summarized in Table 1. The following two observations can be drawn: (1) with the same amount of training data (from $20\%$ to $100\%$), our method consistently outperforms the baseline model; and (2) the performance gains are larger when the amount of training data is smaller. This phenomenon is understandable, as limited training data makes models more susceptible to overfitting (see Figure 1), which leads to poor generative performance. For example, on the $20\%$ FFHQ training set, our method yields a $15.88\%$ improvement in FID compared with the baseline EDM-NCSN++ method. However, as the training data increases, the performance gain diminishes to $1.16\%$. We further evaluate our method on the high-resolution

benchmark LSUN-Church, with results reported in Table 5 in Appendix C.3. The conclusions are consistent with those drawn from the low-resolution benchmarks.

## 4.3 EXPERIMENTS ON FEW-SHOT GENERATION

In practice, it is often impossible to collect a large-scale dataset for specific images of interest. To address this few-shot image generation problem, researchers recently exploit few-shot learning (Gharoun et al., 2024; Wang et al., 2020a) in the setting of image generation, including LD-Diffusion (Zhang et al., 2025), LPDM-8 (Wang et al., 2023a), FreezeD (Mo et al., 2020), Transfer-GAN (Wang et al., 2018b), MineGAN (Wang et al., 2020b), and DiffAugment (Zhao et al., 2020). We compare these transfer learning approaches with our data-efficient training scheme. Note that these diffusion-based transfer learning methods start from a pre-trained EDM-NCSN++ (Karras et al., 2022) model on the FFHQ dataset, while these GAN-based methods start from a pre-trained StyleGAN-v2 (Karras et al., 2019) model on the same dataset. Our comparison experiments are conducted on the 100-shot datasets (Obama, Grumpy Cat, and Panda) (Zhao et al., 2020), and AnimalFace (160 cats and 389 dogs) (Si & Zhu, 2011). The results in Table 2 show that WILD-Diffusion achieves consistent gains on all datasets, with or without pre-training. For example, our method achieves the lowest FID score of $34.52$ (representing an improvement of at least $7\%$) on the 100-shot Obama dataset when trained from scratch.

Table 2: The FID results on few-shot generation. Following the setting used in (Zhao et al., 2020), we calculate the FID with $5k$ samples and the training dataset is adopted as the reference distribution. All transfer learning methods have their pre-trainings from the FFHQ dataset. The numerical results of the baseline methods are quoted from their papers. We highlight the best results in **bold**.

| Methods | Architecture | Pre-training? | 100-shot | | | Animal-Face | |
| --- | --- | --- | --- | --- | --- | --- | --- |
| | | | Obama | Grumpy | Panda | Cat | Dog |
| StyleGAN-v2 (Karras et al., 2019) | GAN | No | 80.20 | 48.90 | 34.27 | 71.71 | 130.19 |
| EDM-NCSN++ (Karras et al., 2022) | Diffusion | No | 37.10 | 29.94 | 10.81 | 36.88 | 57.14 |
| MineGAN (Wang et al., 2020b) | GAN | Yes | 50.63 | 35.54 | 14.84 | 54.45 | 93.03 |
| TransferGAN (Wang et al., 2018b) | GAN | Yes | 48.73 | 34.06 | 23.20 | 52.61 | 82.38 |
| FreezeD (Mo et al., 2020) | GAN | Yes | 41.87 | 31.22 | 17.95 | 47.70 | 70.46 |
| LPDM-8 (Wang et al., 2023a) | Diffusion | Yes | 14.27 | 14.56 | 5.13 | 14.92 | 15.95 |
| LD-Diffusion (Zhang et al., 2025) | Diffusion | Yes | 13.00 | 13.31 | 4.70 | **12.77** | 12.48 |
| **WILD-Diffusion (ours)** | Diffusion | Yes | **12.54** | **12.83** | **4.66** | 12.93 | **12.21** |
| DiffAugment (Zhao et al., 2020) | GAN | No | 46.87 | 27.08 | 12.06 | 42.44 | 58.85 |
| Patch Diffusion (Wang et al., 2023a) | Diffusion | No | 41.47 | 30.89 | 13.25 | 43.71 | 72.17 |
| **WILD-Diffusion (ours)** | Diffusion | No | **34.52** | **26.33** | **9.96** | **34.21** | **53.18** |

## 4.4 ABLATION EXPERIMENTS

Given that our proposed method incorporates the "Wasserstein distance" (Eq.(2)), it is natural to compare with other distributional divergences that are commonly used in distributionally robust optimization (DRO). To this end, we perform an ablation study by replacing the Wasserstein distance with alternative divergences: (1) KL-divergence, (2) $\chi^2$-divergence, and (3) $\alpha$-divergence (see detailed definition in Appendix D). The results are summarized in Figure 11 in Appendix C.6, which demonstrate that our method achieves superior performance compared to these alternatives. Furthermore, as our method can be regarded as a novel data augment method with theoretical guarantee, we perform the ablation experiments comparing it against representative augmentation techniques, including Mixup (Zhang et al., 2018), CutMix (Yun et al., 2019), and CutOut (DeVries & Taylor, 2017). The results are presented in Table 8 in Appendix C.6, which show that our method achieves better performance than other methods.

## 5 CONCLUSION

In this paper, we introduced WILD-Diffusion, a novel diffusion training framework based on WDRO. Our method dynamically expands the support of the training distribution, which mitigates overfitting and improves generation quality under limited data. We proposed an efficient algorithm with a theoretical convergence guarantee, and extensive experiments demonstrated that WILD-Diffusion can improve state-of-the-art diffusion models across diverse datasets and architectures.

ACKNOWLEDGMENTS

The authors would like to thank the anonymous reviewers for their valuable comments and suggestions. This work was partially supported by the National Key Research and Development Program of China (No. 2021YFA1000900), the National Natural Science Foundation of China (No. 62272432, No. 62432016), and the Natural Science Foundation of Anhui Province (No. 2208085MF163).

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

# A  RELATED WORK

Recent advances in generative modeling have been driven by diffusion models, which have achieved state-of-the-art performance across a wide range of applications. However, their effectiveness in limited data settings remains a major challenge, as models often suffer from overfitting. To address this issue, prior works have explored strategies such as data augmentation and few-shot adaptation. In parallel, the framework of Wasserstein distributionally robust optimization (WDRO) has emerged as a powerful tool for mitigating overfitting by optimizing against worst-case perturbations of data distributions. In this section, we review related work along three directions: diffusion models, generative modeling under limited data, and WDRO.

**Denoising diffusion probabilistic models (DDPM).** In recent years, diffusion models (Ho et al., 2020; Sohl-Dickstein et al., 2015; Karras et al., 2022; Song et al., 2021b) have emerged as a state-of-the-art family of generative models. They work by sequentially corrupting training data with gradually increasing levels of noise (i.e., *the forward process*), and then learning to reverse this corruption to construct a generative model of the data (i.e., *the reverse process*). Current research on diffusion models has primarily focused on two main formulations: denoising diffusion probabilistic models (DDPM)(Ho et al., 2020; Nichol & Dhariwal, 2021) and score-based stochastic differential equations (Score SDEs)(Song et al., 2021b; Karras et al., 2022) (where score-based generative models (SGMs) (Song & Ermon, 2019; 2020) can be viewed as their discrete counterparts). Given a data point $\mathbf{x}(0) \sim p_{\text{data}}$, the *forward process* generates a sequence of random variables $\{\mathbf{x}(1), \ldots, \mathbf{x}(T)\}$ with the transition kernel $p(\mathbf{x}(t) \mid \mathbf{x}(t-1))$ for all timesetp $t \in \{0, 1, \ldots, T\}$. A common choice for the transition kernel is Gaussian kernel (Yang et al., 2023b), i.e., $p(\mathbf{x}(t) \mid \mathbf{x}(t-1)) = \mathcal{N}(\mathbf{x}(t); \sqrt{1-\beta_t}\mathbf{x}(t-1), \beta_t\mathbf{I})$, where $\beta_t \in (0,1)$ is a sequence of positive noise scales. Following Sohl-Dickstein et al. (2015); Ho et al. (2020), with setting $\alpha_t := 1 - \beta_t$ and $\bar{\alpha}_t := \prod_{s=0}^t \alpha_s$, we have $p(\mathbf{x}(t) \mid \mathbf{x}(0)) = \mathcal{N}(\mathbf{x}(t); \sqrt{\bar{\alpha}_t}\mathbf{x}(0), (1-\bar{\alpha}_t)\mathbf{I})$. Therefore, we can easily obtain a sample of $\mathbf{x}(t)$ by sampling a Gaussian vector $\boldsymbol{\epsilon} \sim \mathcal{N}(\mathbf{0}, \mathbf{I})$ and applying the transformation $\mathbf{x}(t) = \sqrt{\bar{\alpha}_t}\mathbf{x}(0) + \sqrt{1-\bar{\alpha}_t}\boldsymbol{\epsilon}$. Since the noise scales $\bar{\alpha}$ are prescribed (Song et al., 2021b), so that $\mathbf{x}(T)$ is almost Gaussian in distribution, i.e., $\mathbf{x}(T) \sim \mathcal{N}(\mathbf{0}, \mathbf{I})$. The *reverse process* is a variational Markov chain and parameterized with $p_\theta(\mathbf{x}(t-1) \mid \mathbf{x}(t)) = \mathcal{N}(\mathbf{x}(t-1); \frac{1}{\sqrt{1-\beta_t}}(\mathbf{x}(t) + \beta_t\mathbf{s}_\theta(\mathbf{x}(t), t)), \beta_t\mathbf{I})$. Thus, the loss takes the following form (see Song et al. (2021b) for details):

$$\ell(\theta, \mathbf{x}, t) = \lambda(t)\beta_t^2 \cdot \|\mathbf{s}_\theta(\mathbf{x}, t) - \nabla_\mathbf{x} \log p_t(\mathbf{x})\|_2^2 \tag{15}$$

where $\lambda(t)$ is a positive weighting function (Yang et al., 2023b).

**Generative models with limited data.** Prior to the rise of diffusion models, a large body of work studied training schemes for generative models in limited data settings, primarily in the context of GANs (Abdollahzadeh et al., 2023). A significant challenge in this scenario is "overfitting"(Karras et al., 2020; Liu et al., 2021), where the model may memorize the training data (Li et al., 2020; Ojha et al., 2021) and reproduce training examples rather than learn the real data distribution (Zhao et al., 2022). Moreover, under limited data regimes, generative models are more prone to mode collapse (Tran et al., 2021), i.e., the models learn only a limited set of modes and fail to capture other modes of the data distribution, resulting in limited diversity in generated samples (Yu et al., 2022). Various strategies have been proposed to mitigate this phenomenon, primarily focusing on data augmentation (Zhang et al., 2020; Zhao et al., 2021; Karras et al., 2020; Chen et al., 2021; Wang et al., 2023b), which increases the quantity and diversity of the training data. For example, the ADA method (Karras et al., 2020) applies an adaptive augmentation strategy (i.e., with augmentation probability $p < 1$) in the limited data setting to prevent information leakage. Meanwhile, recent works have also begun exploring the few shot adaptation of diffusion models (Lu et al., 2023; Ruiz et al., 2023; Zhang et al., 2025). For instance, DreamBooth (Ruiz et al., 2023) finetunes a pretrained text-to-image model on a few images of a specific subject and introduces a special identifier token in the prompt, enabling the finetuned model to generate diverse images that preserve the subject's identity. However, these works do not fully explore training diffusion models from scratch under limited data, and they differ drastically from our proposed method.

**Wasserstein distributionally robust optimization (WDRO).** WDRO (Rahimian & Mehrotra, 2019a; Wang et al., 2025b) is an effective optimization framework for learning and decision-making under uncertainty (Wozabal, 2014; Rahimian & Mehrotra, 2022; Kuhn et al., 2019; Huang et al., 2025a). The core idea of WDRO is to optimize the worst-case expected loss over a Wasserstein

uncertainty set (also known as an ambiguity set) of plausible distributions, rather than a single empirical distribution (Rahimian & Mehrotra, 2019a). Previous approaches to distributional robustness have considered finite-dimensional parametrizations for the uncertainty set, such as constraint sets for moments, support, or directional deviations (Chen et al., 2007; Delage & Ye, 2010; Goh & Sim, 2010), as well as non-parametric distances for probability measures, such as $f$-divergences (e.g., $\chi^2$ divergence, $\alpha$-divergence, and Kullback-Leibler divergence) (Ben-Tal et al., 2013; Duchi et al., 2021; Namkoong & Duchi, 2016) and Wasserstein distances (Blanchet et al., 2019a;b; Mohajerin Esfahani & Kuhn, 2018; Gao & Kleywegt, 2023; Huang et al., 2022). WDRO has been successfully applied to numerous problems in machine learning, including (semi-)supervised learning (Blanchet & Kang, 2020; Chen & Paschalidis, 2018), adversarial training (Levine & Feizi, 2020; Najafi et al., 2019; Sinha et al., 2018; Staib & Jegelka, 2017; Liu et al., 2025), reinforcement learning (Liu et al., 2022; Abdullah et al., 2019), and transfer learning (Volpi et al., 2018; Lee & Raginsky, 2018). Recent work has also investigated the incorporation of DRO into diffusion models. For instance, Wang et al. (2025a) employ DRO to mitigate the distribution mismatch that arises between the training and sampling procedures. In contrast, Our work differs substantially in both problem setting and DRO formulation: we focus on limited data diffusion training, and we design a WDRO method (i.e., implemented via a "Bi-level Interval Update" strategy) on the original data distribution to expand support and mitigate overfitting.

## B PROOF

In this section, we provide a detailed proof of the convergence result in Section 3.2. In Section B.1, we establish an upper bound on the worst-case objective (2) (i.e., Lemma 3.5). In Section B.2, we present several auxiliary lemmas that are directly used in the proof of Theorem 3.6. Finally, in Section B.3, we establish the convergence result of the WILD-Diffusion algorithm (Theorem 3.6). In addition, we provide the convergence result from Lee et al. (2022) for comparison in Section B.4.

### B.1 PROOF OF LEMMA 3.5

**Lemma B.1** *Under Assumption 3.3, for any fixed $\tau > 0$, the following inequality holds with probability at least $1 - e^{-\tau}$, uniformly over all $\rho \geq 0$ and $\gamma \geq 0$*

$$\sup_{p:\mathcal{W}_{\mathbf{c}}(p,p_{data}) \leq \rho} \mathbb{E}_p[\ell(\theta;\mathbf{x},t)] \leq \gamma\rho + \mathbb{E}_{\hat{p}_n}[\phi_\gamma(\theta;\mathbf{x},t)] + O(\sqrt{\tfrac{\tau}{n}}). \tag{16}$$

*Proof* The proof follows (Sinha et al., 2018). For any data distribution $p_{\text{data}}$ and $\rho > 0$, the following duality result holds for problem (2):

$$\sup_{p:\mathcal{W}_{\mathbf{c}}(p,p_{\text{data}}) \leq \rho} \mathbb{E}_p[\ell(\theta;\mathbf{x},t)] = \inf_{\gamma \geq 0} \left\{ \gamma\rho + \mathbb{E}_{p_{\text{data}}}[\phi_\gamma(\theta;\mathbf{x},t)] \right\}. \tag{17}$$

From the above duality result (17), for all $\rho > 0$, data distributions $p_{\text{data}}$, and $\gamma > 0$, we have

$$\sup_{p:\mathcal{W}_{\mathbf{c}}(p,p_{\text{data}}) \leq \rho} \mathbb{E}_p[\ell(\theta;\mathbf{x},t)] \leq \gamma\rho + \mathbb{E}_{p_{\text{data}}}[\phi_\gamma(\theta;\mathbf{x},t)]. \tag{18}$$

Let $\delta_{\mathbf{x}}$ denote the point mass at $\mathbf{x}$. We first present the empirical result for Eq. (9a):

$$\underset{\theta}{\text{minimize}} \left\{ \mathcal{L}_n(\theta) := \sup_p \left\{ \mathbb{E}_p[\ell(\theta;\mathbf{x},t)] - \gamma\mathcal{W}_{\mathbf{c}}(p,\hat{p}_n) \right\} = \mathbb{E}_{\hat{p}_n}\left[ \phi_\gamma(\theta;\mathbf{x},t) \right] \right\}, \tag{19}$$

where $\hat{p}_n = \frac{1}{n}\sum_{i=1}^n \delta_{\mathbf{x}_i}$ denotes the empirical distribution of the samples $\mathbf{x}_{1:n}$. Next, we show that $\mathbb{E}_{\hat{p}_n}[\phi_\gamma(\theta;\mathbf{x},t)]$ concentrates around its population counterpart at the standard rate (Boucheron et al., 2005).

Since we assume that the loss function $\ell(\theta;\mathbf{x},t)$ is uniformly bounded by $\varepsilon$ in Assumption 3.3, i.e., $|\ell(\theta;\mathbf{x},t)| \leq \varepsilon$. Together with the definition of the surrogate loss, we have that

$$-\varepsilon \leq \ell(\theta;\mathbf{x},t) \leq \phi_\gamma(\theta;\mathbf{x},t) \leq \sup_{\mathbf{x}} \left\{ \ell(\theta;\mathbf{x},t) \right\} \leq \varepsilon,$$

and hence $|\phi_\gamma(\theta;\mathbf{x},t)| \leq \varepsilon$. Thus, the functional $\theta \mapsto \mathcal{L}_n(\theta)$ satisfies the bounded differences (Boucheron et al., 2005).

Note that our bound relies on the usual covering numbers for the model class $\ell(\theta; \cdot) : \theta \in \Theta_1$ as a measure of complexity (Wellner et al., 2013), where $\Theta_1$ denotes the parameter space. Recall the definition of covering numbers: for a set $V$, a collection $\{v_1, \ldots, v_N\}$ is an $\epsilon$-cover of $V$ in norm $\| \cdot \|$ if for each $v \in V$, there exists $v_i$ such that $\|v - v_i\| \leq \epsilon$. Then the covering number of $V$ with respect to $\| \cdot \|$ is

$$N(V, \epsilon, \| \cdot \|) := \inf\{N \in \mathbb{N} \mid \text{there exists an } \epsilon\text{-cover of } V \text{ with respect to } \| \cdot \|\}.$$

For our problem, let $\mathcal{L} := \ell(\theta; \cdot) : \theta \in \Theta_1$ denote the loss function class equipped with the $L_\infty(\mathcal{X})$ norm, i.e.,

$$\|\ell\|_{L_\infty} := \sup_{\mathbf{x} \in \mathcal{X}} |\ell(\mathbf{x})|, \quad \ell \in \mathcal{L}$$

therefore the covering number of $\mathcal{L}$ is $N(\mathcal{L}, \varepsilon\epsilon, \| \cdot \|_{L_\infty})$.

By applying standard results on Rademacher complexity (Bartlett & Mendelson, 2002) and entropy integrals (Wellner et al., 2013), we have that for any fixed $\tau > 0$, the following inequality holds with probability at least $1 - e^\tau$,

$$\mathbb{E}_{p_{\text{data}}}[\phi_\gamma(\theta; \mathbf{x}, t)] \leq \mathbb{E}_{\hat{p}_n}[\phi_\gamma(\theta; \mathbf{x}, t)] + b_1 \gamma \sqrt{\frac{\varepsilon}{n}} \int_0^1 \sqrt{\log N(\mathcal{L}, \varepsilon\epsilon, \| \cdot \|_{L_\infty})} \, d\epsilon + b_2 \, \varepsilon \sqrt{\frac{\tau}{n}}, \quad (20)$$

where $b_1, b_2 > 0$ are absolute constants.

Substituting Eq. (22) into Eq. (18):

$$\sup_{p: \mathcal{W}_\mathbf{c}(p, p_{\text{data}}) \leq \rho} \mathbb{E}_p[\ell(\theta; \mathbf{x}, t)] \leq \gamma\rho + \mathbb{E}_{\hat{p}_n}[\phi_\gamma(\theta; \mathbf{x}, t)] + O\left(\sqrt{\frac{\tau}{n}}\right).$$

$\square$

## B.2 Auxiliary Lemmas

For analytical convenience, we consider the following discretization and approximation of Eq. (5), which can be expressed as

$$\mathbf{x}_{(i+1)h} = \mathbf{x}_{ih} - \int_{ih}^{(i+1)h} \left[\mathbf{f}(\mathbf{x}_{ih}, T - t) - \tfrac{1}{2}g(T - t)^2 \cdot \mathbf{s}_\theta(\mathbf{x}_{ih}, T - ih)\right] dt, \quad (\mathbb{D})$$

where $h$ denotes the step size with $T = kh$ (and $k$ is the number of steps), and time is reversed such that $t$ in the reverse process corresponds to $(T - t)$ in the forward process. Following Lee et al. (2022), our proof method is to construct a "bad set", which is formalized in lemma B.2. Specifically, we define a bad set $B_k$ as the set of $\mathbf{x}_k$ for which the *worst-case error* is large (see Eq. 24). Let $q_k$ denote the discretized process ($\mathbb{D}$) with the estimated score, and $\bar{q}_k$ denote the discretized process ($\mathbb{D}$) that also uses the estimated score except in $B_k$. The following lemma formalizes this construction and provides the key bound needed for our analysis.

**Lemma B.2** *Let $(\Omega, \mathcal{F}, \mathbb{P})$ be a probability space and $\{\mathcal{F}_k\}$ a filtration of $\mathcal{F}$. Suppose $\mathbf{x}_k \sim p_k$, $z_k \sim q_k$, and $\bar{z}_k \sim \bar{q}_k$ are $\mathcal{F}_k$-adapted stochastic processes taking values in $\Omega$. Assume further that if $z_i \in B_i^c$ for all $1 \leq i \leq k - 1$, then $z_k = \bar{z}_k$. Under these conditions, the following results hold*

$$D_{\text{TV}}(q_k, \bar{q}_k) \leq \sum_{i=0}^{k-1} \left(\chi^2(\bar{q}_i \,\|\, p_i) + 1\right)^{1/2} \delta_k^{1/2}, \quad (21)$$

$$D_{\text{TV}}(q_k, p_k) \leq \chi^2(\bar{q}_k \,\|\, p_k)^{1/2} + \sum_{i=0}^{k-1} \left(\chi^2(\bar{q}_i \,\|\, p_i) + 1\right)^{1/2} \delta_k^{1/2}, \quad (22)$$

*where $\delta_k$ satisfies $\mathbb{P}(z_k \in B_k) \leq \delta_k$ for every $k \in \mathbb{N}$.*

Note that $\chi^2(\cdot | \cdot)$ denotes the $\chi^2$-divergence, and its detailed definition is provided in Section D.

*Proof*  By the definition of the total variation distance (Section D), we obtain

$$D_{TV}(q_k, \bar{q}_k) = \mathbb{P}(z_i \neq \bar{z}_i)$$

$$\leq \mathbb{P}\left(\bigcup_{i=0}^{k-1} \{z_i \in B_i\}\right) = \mathbb{P}\left(\bigcup_{i=0}^{k-1} \{\bar{z}_i \in B_i\}\right)$$

$$\leq \sum_{i=0}^{k-1} \mathbb{P}(\bar{z}_i \in B_i) = \sum_{i=0}^{k-1} \mathbb{E}_{q_i} \mathbb{I}_{B_i}$$

$$\leq \sum_{i=0}^{k-1} \left(\mathbb{E}_{p_i}\left(\frac{\bar{q}_i}{p_i}\right)^2\right)^{1/2} (\mathbb{E}_{p_i} \mathbb{I}_{B_i})^{1/2}$$

$$= \sum_{i=0}^{k-1} \left(\chi^2(\bar{q}_i \,\|\, p_i) + 1\right)^{1/2} \delta_i^{1/2}.$$

The second inequality (22) then follows from the triangle inequality and Cauchy–Schwarz:

$$D_{TV}(q_k, p_k) \leq D_{TV}(p_k, \bar{q}_k) + D_{TV}(\bar{q}_k, q_k)$$

$$\leq \chi^2(\bar{q}_k \,\|\, p_k)^{1/2} + D_{TV}(\bar{q}_k, q_k).$$

$\square$

Notice that $\chi^2$ convergence bounds directly yield bounds on the total variation distance between the real distribution $p_{\text{data}}$ and the sampling distribution $q_0$ (with $k = 0$). We therefore recall the convergence result of Lee et al. (2022) as follows.

**Lemma B.3** *( Lee et al. (2022, Theorem 4.3)) Let $p : \mathbb{R}^d \to \mathbb{R}$ be a probability density satisfying Assumption 3.2, and let $\mathbf{s}_\theta(\mathbf{x}, t) : \mathbb{R}^d \times [0, T] \to \mathbb{R}^d$ be a score estimator with error bounded in $L^\infty$ norm for each $t \in [0, T]$:*

$$\|\nabla \ln p_t(\mathbf{x}) - \mathbf{s}_\theta(\mathbf{x}, t)\|_\infty = \max_{\mathbf{x} \in \mathbb{R}^d} \|\nabla \ln p_t(\mathbf{x}) - \mathbf{s}_\theta(\mathbf{x}, t)\| \leq \varepsilon_1.$$

*Let $T = O\left(\max\{1, \log(C_{\text{lS}}d)\}\right)$ and $h = \Theta\left(\frac{1}{C_{\text{lS}}(C_{\text{lS}}+d)\max\{L^2, L_s^2\}}\right)$. If $\varepsilon_1 < \frac{1}{128 C_{\text{lS}}}$, then*

$$\chi^2(\bar{q}_0 \,\|\, p_{data}) = \exp\left(-\frac{T}{16 C_{\text{lS}}}\right) \chi^2(q_0 \,\|\, p_{data}) + O(C_{\text{lS}}\varepsilon_1^2) + O\left((L_s^2 + L^2)C_{\text{lS}}h\right). \tag{23}$$

**Lemma B.4** *Suppose that distribution $p$ has log-Sobolev constant at most $C_{\text{lS}}$ and satisfy Assumption 3.2. Then for $T = O(\log(C_{\text{lS}}d))$,*

$$\chi^2(q_0 \,\|\, p_{data}) = O(1).$$

For a detailed proof plsese see (Lee et al., 2022, Lemma E.9).

### B.3  PROOF OF THEOREM 3.6

We first define a sequence of "bad" sets $B_{t \in [0,T]}$ where the *worst-case* error in the score estimate is large,

$$B_t := \left\{\mathbf{x} \in \mathbb{R}^d : \sup_{p:\mathcal{W}_c(p, p_{\text{data}}) \leq \rho} \mathbb{E}_{\mathbf{x} \sim p}\left[\|\mathbf{s}_\theta(\mathbf{x}, T - t) - \nabla \log p_t(\mathbf{x})\|^2\right] > \varepsilon_B\right\}, \tag{24}$$

for some $\varepsilon_B$ to be chosen. Define $t_- := h \lfloor \frac{t}{h} \rfloor$ for all $t \geq 0$. We recall the discretization sampling process ($\mathbb{D}$) and define an interpolated process as

$$\bar{\mathbf{x}}_t = \mathbf{x}_{t_-} - \left[\mathbf{f}(\mathbf{x}_{t_-}, T - t) - \tfrac{1}{2}g(T - t)^2 b(\mathbf{x}_{t_-}, T - t)\right] \mathrm{d}t,$$

where

$$b(\mathbf{x}, t) = \begin{cases} \mathbf{s}_\theta(\mathbf{x}, t), & \mathbf{x} \notin B_t, \\ \nabla \ln p_t(\mathbf{x}), & \mathbf{x} \in B_t. \end{cases}$$

Specifically, we simulate the ODE (5) using the score estimator $\mathbf{s}_\theta$ whenever the point lies in the good set at the "previous" discretization step (i.e., at time $t_-$), and replace it with the true gradient $\nabla \ln p_t$ otherwise. Note that this interpolated process is introduced purely for analysis, since $\nabla \ln p_t$ is not available in practice.

Then, applying Chebyshev's inequality (Knuth, 1997) and Lemma 3.5 to the Eq. (24), we obtain

$$\mathbb{P}(B_t) \leq \frac{\gamma\rho + \mathbb{E}_{\hat{p}_n}[\phi_\gamma(\theta;\mathbf{x},t)] + O(\sqrt{\frac{1}{n}})}{\varepsilon_B^2}. \tag{25}$$

Recall Lemma B.3 (Eq. (22)), we have

$$\chi^2(\bar{q}_0 \,\|\, p_{\text{data}}) = \exp\left(-\frac{T}{16C_{\text{lS}}}\right)\chi^2(q_0 \,\|\, p_{\text{data}}) + O(C_{\text{lS}}\varepsilon_B^2) + O\left((L_s^2 + L^2)C_{\text{lS}}h\right). \tag{26}$$

To ensure that this quantity is bounded by $\varepsilon_\chi^2$, it suffices to require

$$\exp\left(-\frac{T}{16C_{\text{lS}}}\right)\chi^2(q_0 \,\|\, p_{\text{data}}) \leq \frac{\varepsilon_\chi^2}{2},$$

$$C_{\text{lS}}\varepsilon_B^2 \leq \frac{\varepsilon_\chi^2}{4},$$

$$(L_s^2 + L^2 d)C_{\text{lS}}h \leq \frac{\varepsilon_\chi^2}{4}.$$

Consequently, we obtain

$$T \geq 32C_{\text{lS}}\log\left(\frac{\varepsilon_\chi^2}{2\chi^2(q_0 \,\|\, p_{\text{data}})}\right),$$

$$h \leq \frac{\varepsilon_\chi^2}{4C_{\text{lS}}(L_s^2 + L^2 d)},$$

$$\varepsilon_B \leq \sqrt{\frac{\varepsilon_\chi^2}{4C_{\text{lS}}}}.$$

To satisfy the condition in Lemma B.3, we choose $h = \Theta\left(\frac{\varepsilon_\chi^2}{C_{\text{lS}}(C_{\text{lS}}+d)\max\{L^2, L_s^2\}}\right)$. Note that Eq. (26) also satisfies $\leq \varepsilon_\chi^2$ since $C_{\text{lS}} > 1$. Furthermore, by Lemma B.3 (Eq. (21)), we have

$$\begin{aligned}
\mathrm{D}_{\text{TV}}(q_0, \bar{q}_0) &\leq \sum_{i=0}^{k-1}\left(1 + \chi^2(q_{ih} \,\|\, p_{\text{data}})\right)^{1/2}\mathbb{P}(B_{ih})^{1/2} \\
&\leq \left(\sum_{i=0}^{k-1}\exp\left(-\frac{ih}{32C_{\text{lS}}}\right)\chi^2(q_0 \,\|\, p_{\text{data}})^{1/2} + O(1)\right)\left(\frac{\gamma\rho + \mathbb{E}_{\hat{p}_n}[\phi_\gamma(\theta;\mathbf{x},t)] + O(\sqrt{\frac{1}{n}})}{\varepsilon_B^2}\right)^{1/2} \\
&\leq \left(\sum_{i=0}^{\infty}\exp\left(-\frac{ih}{32C_{\text{lS}}}\right)\chi^2(q_0 \,\|\, p_{\text{data}})^{1/2} + O(k)\right)\left(\frac{\gamma\rho + \mathbb{E}_{\hat{p}_n}[\phi_\gamma(\theta;\mathbf{x},t)] + O(\sqrt{\frac{1}{n}})}{\varepsilon_B^2}\right)^{1/2} \\
&\leq \left(\frac{\gamma\rho + \mathbb{E}_{\hat{p}_n}[\phi_\gamma(\theta;\mathbf{x},t)] + O\left(\sqrt{\frac{1}{n}}\right)}{\varepsilon_B^2}\right)^{1/2}\left(\frac{64C_{\text{lS}}}{h}\chi^2(q_0 \,\|\, p_{\text{data}})^{1/2} + O(k)\right) \\
&\leq \left(\frac{\gamma\rho + \mathbb{E}_{\hat{p}_n}[\phi_\gamma(\theta;\mathbf{x},t)] + O\left(\sqrt{\frac{1}{n}}\right)}{\varepsilon_B^2}\right)^{1/2} \cdot O\left(\max\left\{k, \frac{C_{\text{lS}}\chi^2(q_0 \,\|\, p_{\text{data}})^{1/2}}{h}\right\}\right)
\end{aligned}$$

By Lemma B.4, we obtain that $\chi^2(q_0 \,\|\, p_{\text{data}}) = O(1)$ when $T = \Theta(\log(C_{\text{lS}}d))$. Thus, if $T$ is chosen such that

$$T = \Theta\left(\max\left\{\log(C_{\text{lS}}d),\ C_{\text{lS}}\log\left(\frac{2}{\varepsilon_\chi^2}\right)\right\}\right),$$

we have

$$D_{\mathrm{TV}}(q_0, \bar{q}_0) \leq O\left(\sqrt{\gamma\rho + \mathbb{E}_{\hat{p}_n}[\phi_\gamma(\theta; \mathbf{x}, t)] + O\left(\sqrt{\frac{1}{n}}\right)} \cdot \frac{C_{\mathrm{lS}}^{5/2}(C_{\mathrm{lS}}+d)(L^2+L_s^2)\left(1+\log\left(\frac{2}{\varepsilon_\chi^2}\right)\right)}{\varepsilon_\chi^3}\right).$$

Therefore, by Eq. (22), we get

$$\begin{aligned}
D_{\mathrm{TV}}(q_0, p_{\mathrm{data}}) &\leq \chi^2(\bar{q}_0 \,\|\, p_{\mathrm{data}})^{1/2} + D_{\mathrm{TV}}(q_0, \bar{q}_0) \\
&\leq \varepsilon_\chi + D_{\mathrm{TV}}(q_0, \bar{q}_0). \qquad \text{(Using Eq. (26))}
\end{aligned}$$

### B.4 BACKGROUND THEOREMS

For reference, we include the convergence result of the reverse SDE (i.e., Eq. 4) with the estimated score from Lee et al. (2022), and the detailed result is presented in Lemma B.5 below.

**Lemma B.5** *(Lee et al. (2022) Theorem 3.1) Let $p_{data} : \mathbb{R}^d \to \mathbb{R}$ be a probability density satisfying Assumption 3.2, and let $p_t$ be the distribution resulting from evolving the forward SDE according to DDPM with $g = 1$. Suppose furthermore that $\nabla \log p_t$ is L-Lipschitz for every $t \geq 0$, and that each $s_\theta(\cdot, t)$ satisfies Assumption 3.3. Then if*

$$\varepsilon = O\left(\frac{\varepsilon_{\mathrm{TV}}\varepsilon_\chi^3}{(C_{lS}+d)C_{lS}^{5/2}(\max\{L, L_s\})^2 \max\left\{\log(C_{lS}d), C_{lS}\log(1/\varepsilon_\chi^2)\right\}}\right),$$

*running ($\mathbb{D}$) starting from prior distribution for time $T = \Theta\left(\max\left\{\log(C_{lS}d), C_{lS}\log\left(\frac{1}{\varepsilon_\chi}\right)\right\}\right)$ and step size $h = \Theta\left(\frac{\varepsilon_\chi^2}{C_{lS}(C_{lS}+d)(\max\{L, L_s\})^2}\right)$ results in a distribution $q_0$ so that $D_{\mathrm{TV}}(q_0, p_{data}) \leq \varepsilon_\chi^2 + \varepsilon_{\mathrm{TV}}$.*

## C MORE EXPERIMENT RESULTS

In this section, we provide additional experimental results to further validate the effectiveness of our proposed WILD-Diffusion method. We begin with detailed implementation settings in Section C.1. Next, we present sensitivity analyses of key hyper-parameters in Section C.2, followed by supplementary results under limited data settings in Section C.3 and few-shot generation tasks in Section 4.3. Furthermore, in Section C.5, we extend our method to text-to-image generation. Finally, in Section C.6, we conduct ablation studies to examine the contributions of different components in our method.

### C.1 EXPERIMENTAL IMPLEMENTATION DETAILS

We developed our method on top of a widely used codebase EDM (Karras et al., 2022). We implemented and trained our model with PyTorch on a 64-bit Linux machine with 8 NVIDIA A100 (80G) GPUs. As described in the experimental setting in Section 4, our method is built upon three different models: DDPM++ (Song et al., 2021b), NCSN++ (Song et al., 2021b), and ADM (Dhariwal & Nichol, 2021). Specifically, we highlight the architectural differences among these three models, as illustrated in Table 3. In addition, we provide the detailed training configurations in Table 4.

### C.2 EXPERIMENT RESULTS FOR SENSITIVITY OF HYPER-PARAMETER

Following the sensitivity analysis in Section 4.1, we present the FID and computation time across different settings of the interval parameter $m$ on 50% FFHQ datasets. As shown in Figure 3, increasing $m$ reduces the total training time but also degrades generative performance (higher FID), revealing a clear trade-off between efficiency and quality. Considering both training efficiency and generative quality, we set $m = 20$ as the default choice in all experiments.

Additionally, we also perform a sensitivity analysis on the key hyperparameters of WILD-Diffusion, including the number of steps $K$, the step size $\eta$, and the penalty parameter $\gamma$. As shown in Figure 4, we observe that the generation performance is sensitive to the choice of the number of steps $K$, the

Table 3: Details of the network architectures used in this paper.

| Parameter | DDPM++ | NCSN++ | ADM |
|---|---|---|---|
| Resampling filter | Box | Bilinear | Box |
| Noise embedding | Positional | Fourier | Positional |
| Skip connections in encoder | – | Residual | – |
| Skip connections in decoder | – | – | – |
| Residual blocks per resolution | 4 | 4 | 3 |
| Attention resolutions | {16} | {16} | {32, 16, 8} |
| Attention heads | 1 | 1 | 6-9-12 |
| Attention blocks in encoder | 4 | 4 | 9 |
| Attention blocks in decoder | 2 | 2 | 13 |

Table 4: Hyperparameters used for the training runs in Section 4.

| Datasets | Duration (Mimg) | Minibatch size | lr |
|---|---|---|---|
| CIFAR-10 (20% / 100%) | 200 | 1024 | $10e^{-5}/10e^{-4}$ |
| FFHQ & CelebA-HQ | 200 | 512 | $2e^{-4}$ |
| LSUN-Church | 200 | 256 | $1e^{-4}$ |

step size $\eta$, and the penalty parameter $\gamma$. Specifically, too few steps or a very small step size leads to weak perturbations, which reduces the effectiveness of WILD-Diffusion, while overly large values introduce instability and degrade the FID. Similarly, the penalty parameter $\gamma$ controls the trade-off between perturbation strength and stability, where extreme values yield suboptimal results. Based on this analysis, we set the default configuration as $K = 5$, $\eta = 0.01$, and $\gamma = 1$.

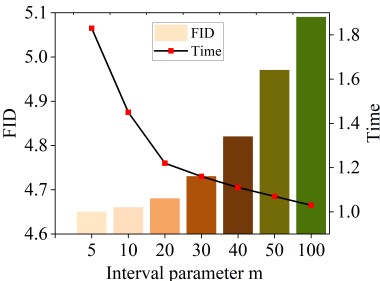

Figure 3: Sensitivity to the interval parameter $m$. Bars show FID (lower is better); the black line shows training time (normalized). Increasing $m$ (less frequent WDRO updates) reduces time but degrades FID, which reveals a trade-off between efficiency and quality.

### C.3 MORE EXPERIMENT RESULTS FOR LIMITED DATA SETTING

In this section, we compare our method with state-of-the-art diffusion approaches on the high-resolution benchmark LSUN-Church ($256 \times 256$). Namely, the baselines include DDPM (Ho et al., 2020), DDIM (Song et al., 2021a), DeepCache (Ma et al., 2024), and EDM-ADM (Karras et al., 2022). The results are summarized in Table 5, and suggest that our method can outperform the baseline models. We further include Figure 5 to separately visualize the off-distribution samples produced by our bi-level update. In addition, Figures 6, 7, 8, and 9 present generative samples from WILD-Diffusion trained on CIFAR-10, FFHQ, CelebA-HQ, and LSUN-Church.

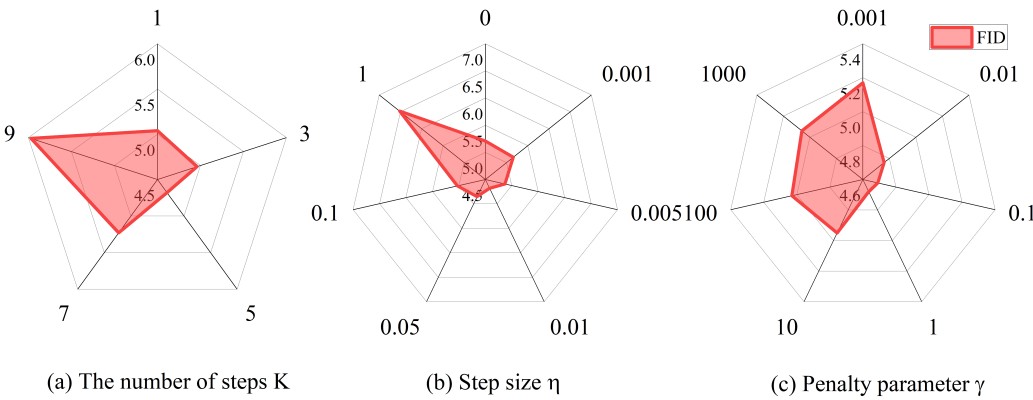

(a) The number of steps K          (b) Step size η          (c) Penalty parameter γ

Figure 4: Sensitivity analysis of WILD-Diffusion with respect to: (a) the number of steps $K$, (b) the step size $\eta$, and (c) the penalty parameter $\gamma$.

Table 5: A comparison of FID between WILD-Diffusion and other diffusion models on the LSUN-Church ($256 \times 256$) dataset. The best results are highlighted in **bold**.

| Methods | Data size | | |
|---|---|---|---|
| | 20% | 50% | 100% |
| DDPM (Ho et al., 2020) | - | - | 7.89 |
| DDIM (Song et al., 2021a) | - | - | 10.58 |
| DeepCache (Ma et al., 2024) | - | - | 11.31 |
| EDM-ADM (Karras et al., 2022) | 7.74 | 5.79 | 4.66 |
| + WILD-Diffusion | **6.98** (-9.82%) | **5.13** (-11.40%) | **4.47** (-4.07%) |

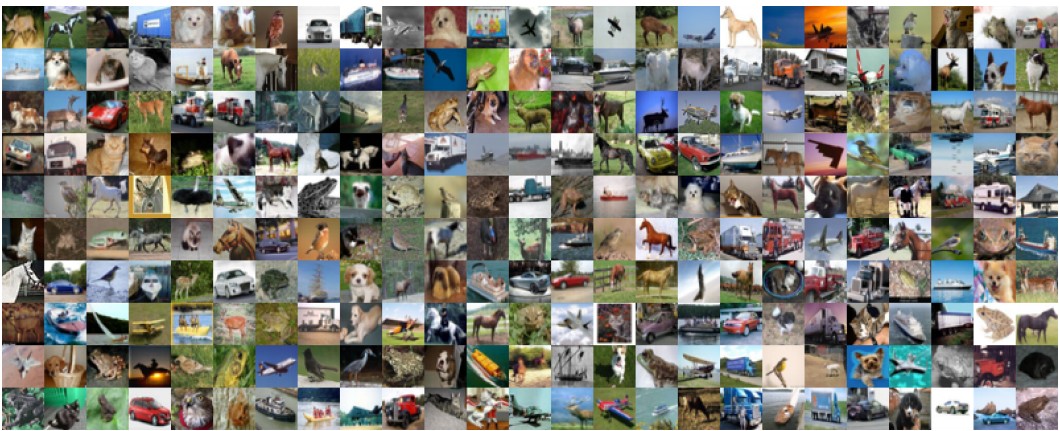

Figure 5: Off-distribution samples generated by the bi-level update.

## C.4  MORE EXPERIMENT RESULTS FOR FEW-SHOT GENERATION

In Table 6, we report the FID results of WILD-Diffusion on the 100-shot, Animal-Face, CelebA-HQ, and LSUN-Cat datasets using a GAN architecture. As described in Section 4, we adopt the pre-trained StyleGAN-v2 (Karras et al., 2019), trained on the FFHQ dataset, as the source model. We compare our method with GAN-based approaches for limited data generation, including Dif-fAugment (Zhao et al., 2020), ADA (Karras et al., 2020), and MAFP (Zhang et al., 2025). The results suggest that our method can achieve the lowest FID scores across all datasets. In addition,

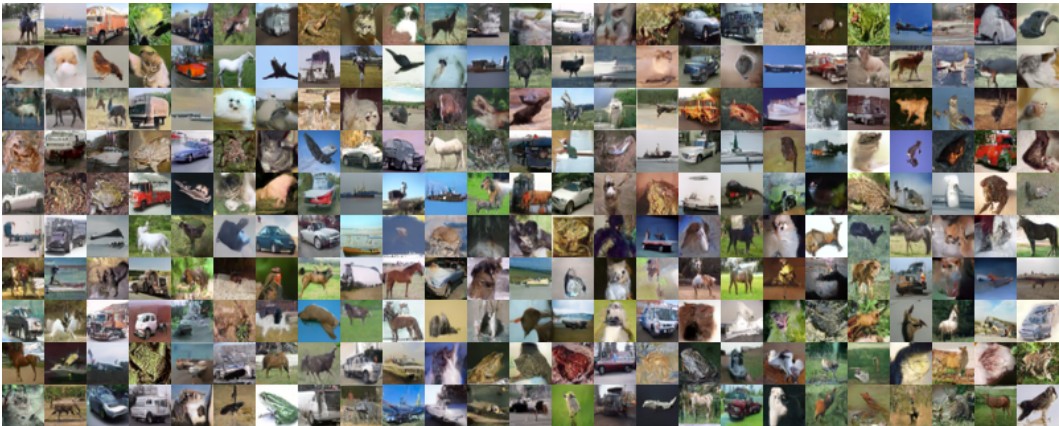

(a) Samples generated on CIFAR-10 ($32 \times 32$) with $20\%$ of the training data. FID = 12.14.

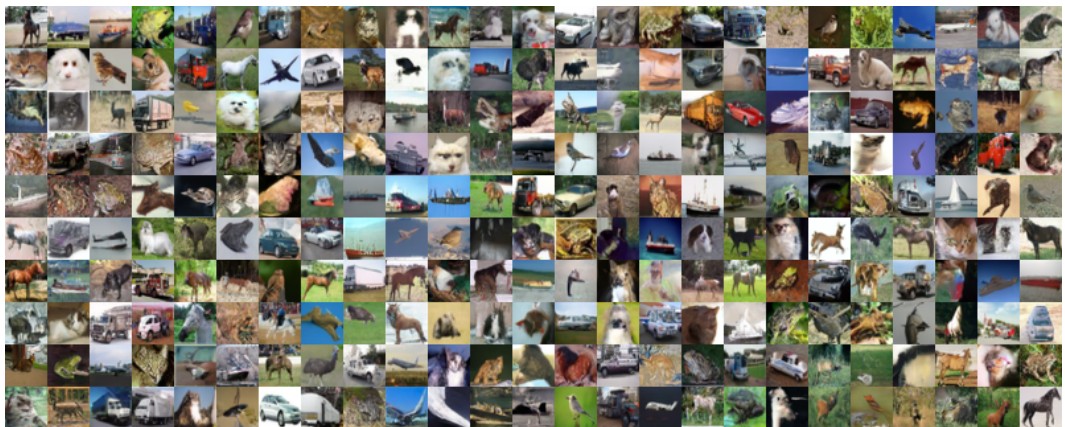

(b) Samples generated on CIFAR-10 ($32 \times 32$) with $50\%$ of the training data. FID = 6.02.

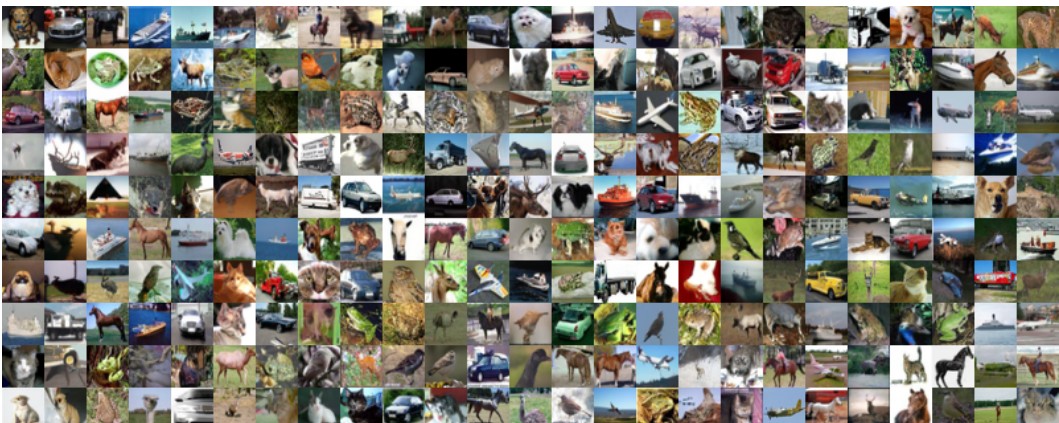

(c) Samples generated on CIFAR-10 ($32 \times 32$) with $100\%$ of the training data. FID = 1.93.

Figure 6: Samples generated on CIFAR-10 ($32 \times 32$) using different proportions of the training data with EDM-DDPM++ (Karras et al., 2022) combined with WILD-Diffusion.

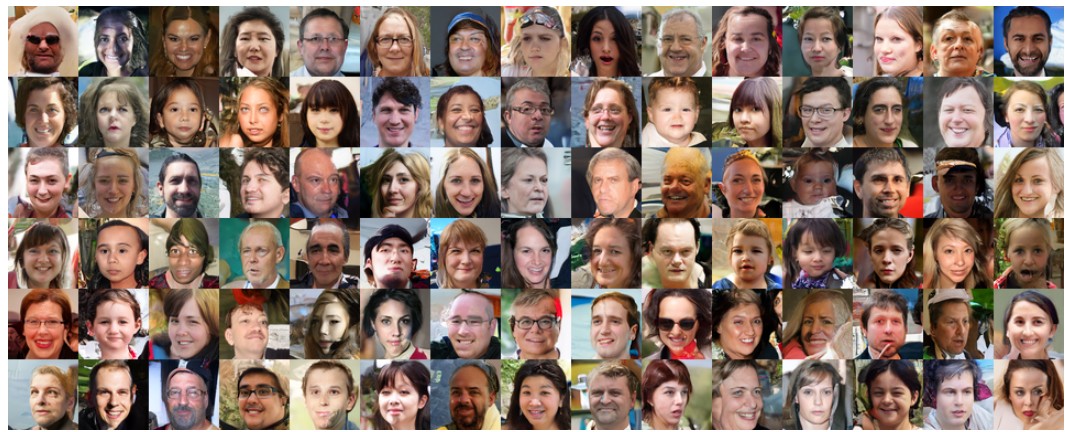

(a) Samples generated on FFHQ ($64 \times 64$) with $20\%$ of the training data. FID = 8.57.

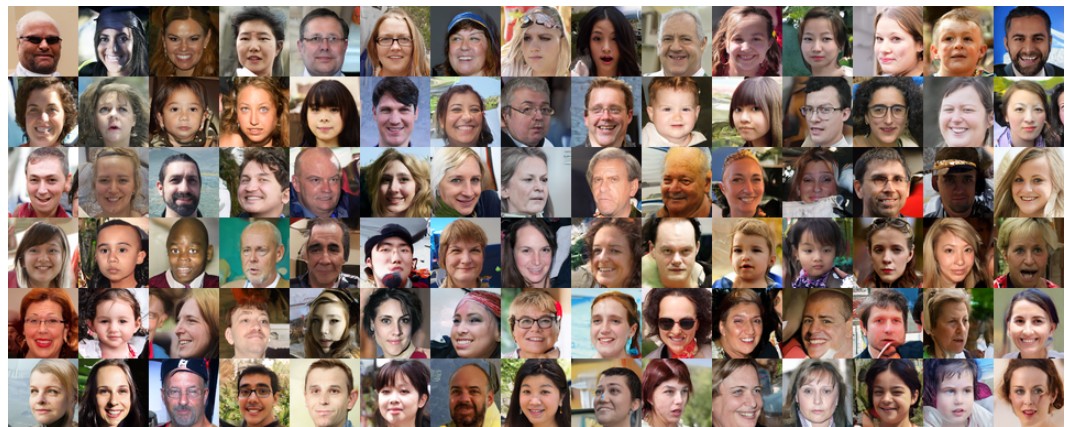

(b) Samples generated on FFHQ ($64 \times 64$) with $50\%$ of the training data. FID = 4.68.

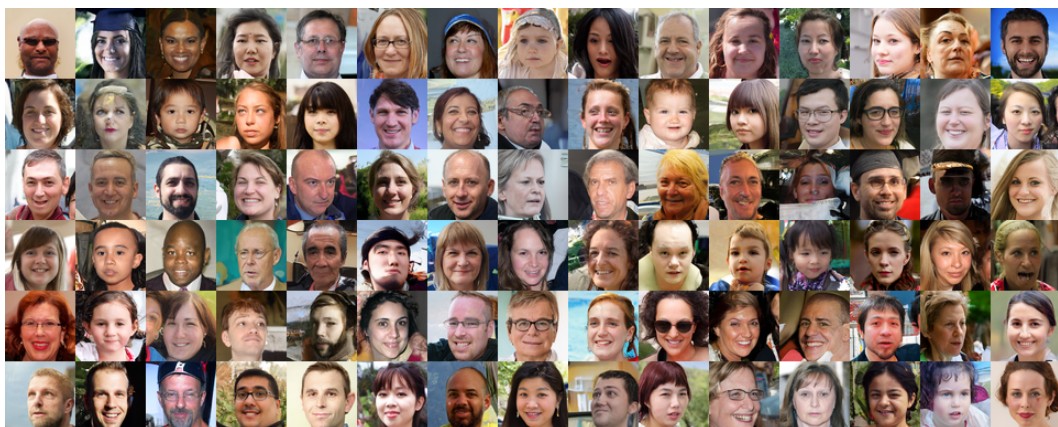

(c) Samples generated on FFHQ ($64 \times 64$) with $100\%$ of the training data. FID = 2.53.

Figure 7: Samples generated on FFHQ ($64 \times 64$) using different proportions of the training data with EDM-DDPM++ (Karras et al., 2022) combined with WILD-Diffusion.

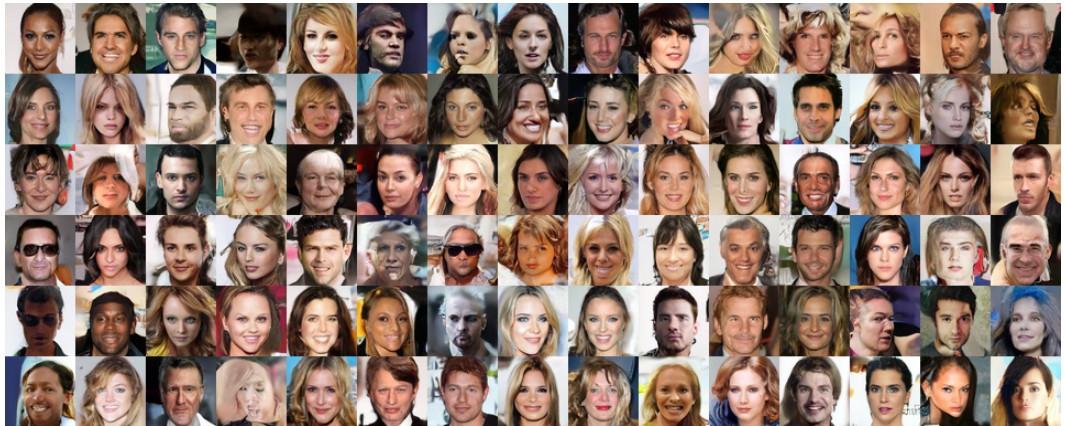

(a) Samples generated on CelebA-HQ ($64 \times 64$) with 20% of the training data. FID = 10.22.

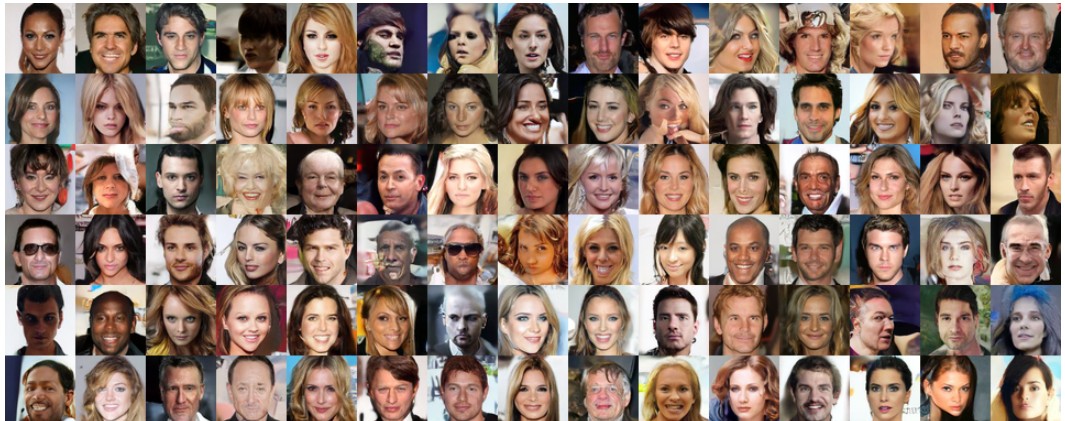

(b) Samples generated on CelebA-HQ ($64 \times 64$) with 50% of the training data. FID = 5.55.

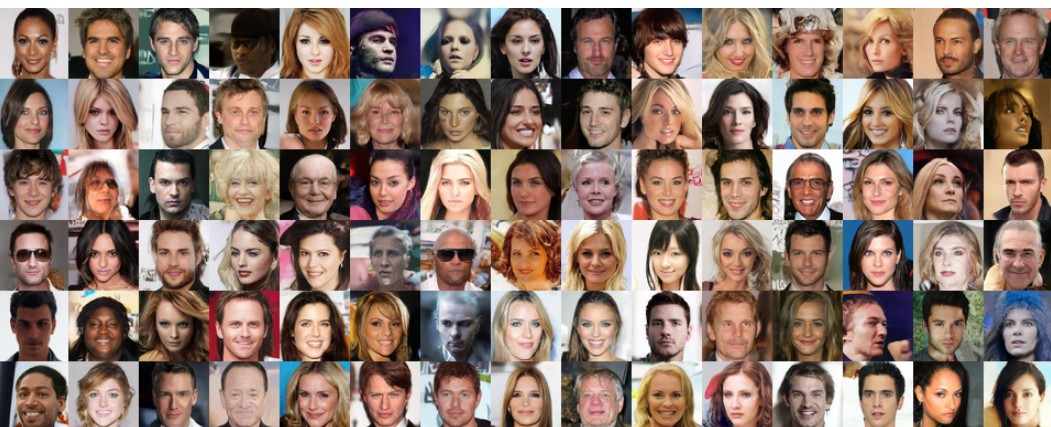

(c) Samples generated on CelebA-HQ ($64 \times 64$) with 100% of the training data. FID = 3.63.

Figure 8: Samples generated on CelebA-HQ ($64 \times 64$) using different proportions of the training data with EDM-DDPM++ (Karras et al., 2022) combined with WILD-Diffusion.

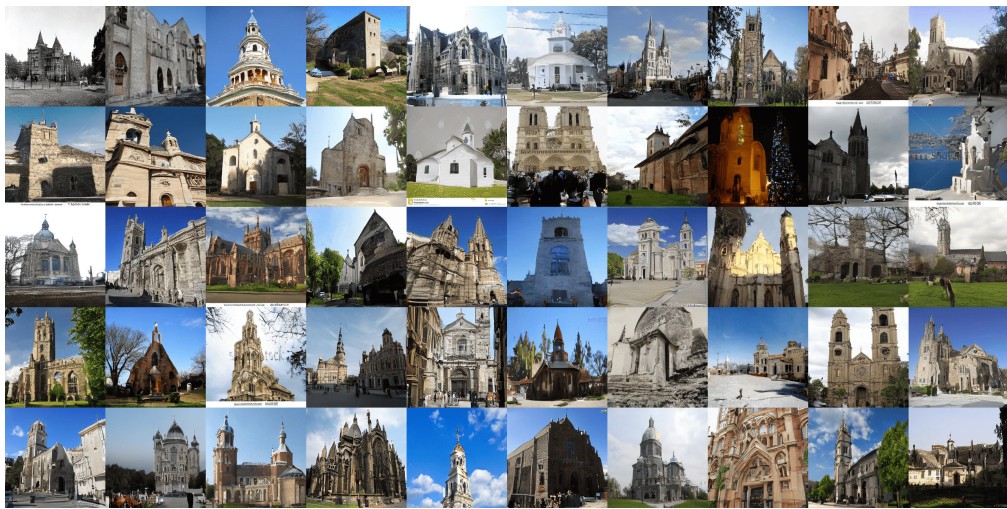

(a) Samples generated on LSUN-Church ($256 \times 256$) with $20\%$ of the training data. FID = 6.98.

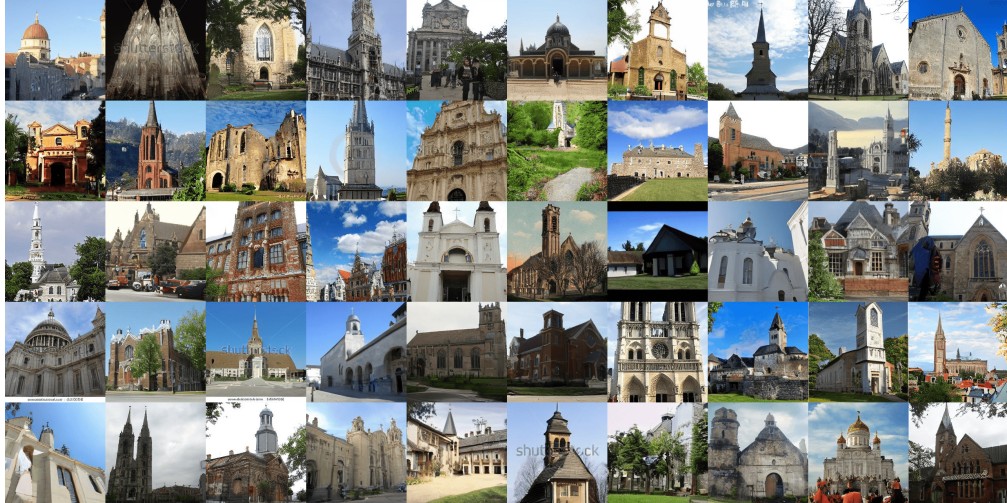

(b) Samples generated on LSUN-Church ($256 \times 256$) with $50\%$ of the training data. FID = 5.13.

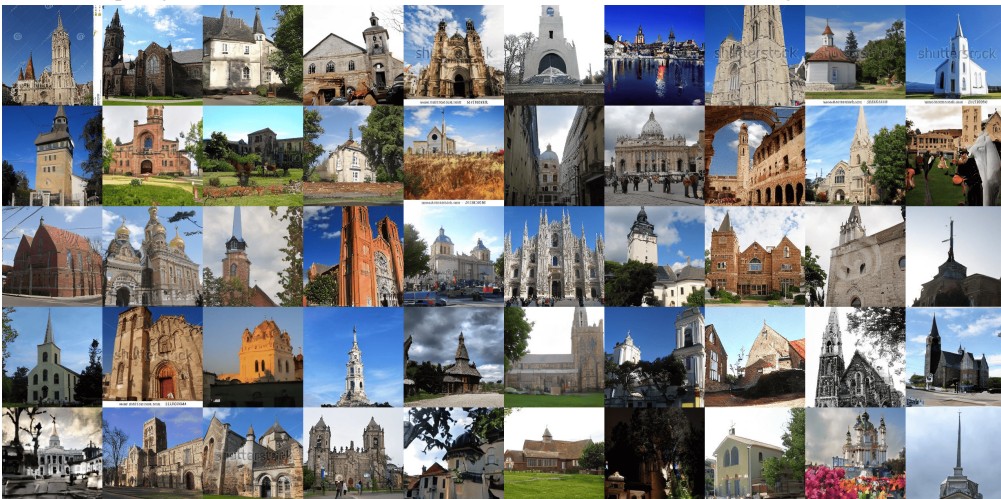

(c) Samples generated on LSUN-Church ($256 \times 256$) with $100\%$ of the training data. FID = 4.47.

Figure 9: Samples generated on LSUN-Church ($256 \times 256$) using different proportions of the training data with EDM-ADM (Karras et al., 2022) combined with WILD-Diffusion.

we provide generative samples from WILD-Diffusion in both pretrained and non-pretrained settings in Figure 10.

Table 6: The FID results on few-shot generation with GAN architecture. Following the setting used in (Zhao et al., 2020), we calculate the FID with $5k$ samples and the training dataset is adopted as the reference distribution. When FFHQ and LSUN-Cat are used as the target datasets, the number of target domain images is $2k$. The numerical results of the baseline methods are quoted from their papers. We highlight the best results in **bold**.

| Methods | FFHQ → 100-shot | | | FFHQ → Animal-Face | | CelebA-HQ → FFHQ | FFHQ → LSUN-Cat |
|---|---|---|---|---|---|---|---|
| | Obama | Grumpy | Panda | Cat | Dog | | |
| DiffAugment (Zhao et al., 2020) | 46.87 | 27.08 | 12.06 | 42.44 | 58.85 | 11.20 | 20.18 |
| ADA (Karras et al., 2020) | 45.69 | 26.62 | 12.90 | 40.77 | 56.83 | 10.08 | 19.34 |
| MAFP (Zhang et al., 2025) | 41.13 | 25.87 | 10.93 | 38.69 | 54.15 | 9.67 | 17.93 |
| Ours | **40.02** | **24.97** | **10.52** | **37.66** | **54.03** | **8.53** | **16.28** |

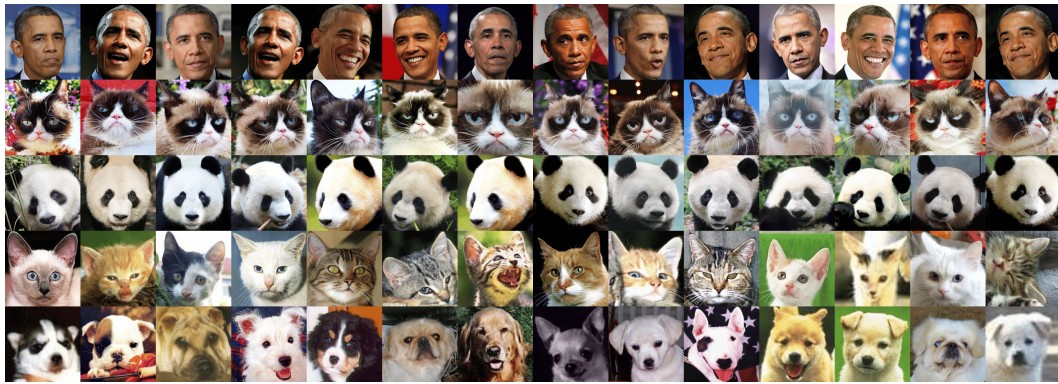

(a) Few-shot generation results of our method without pretraining.

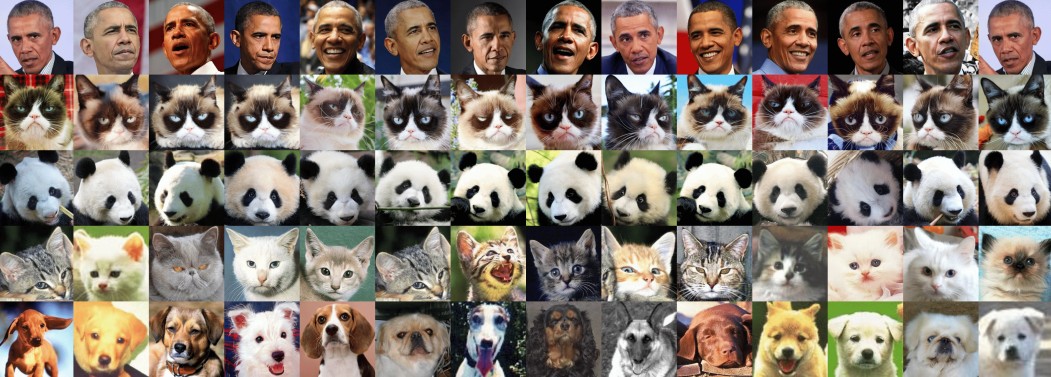

(b) Few-shot generation results of our method with pretraining.

Figure 10: Few-shot image generation results of our method on 100-shot and Animal-Face datasets, shown in both (a) non-pretrained and (b) pretrained settings.

## C.5 EXPERIMENTAL RESULTS FOR TEXT-TO-IMAGE

In this section, we evaluate whether our method generalizes to conditional diffusion models by testing WILD-Diffusion on a standard text-to-image personalization task. We adopt DreamBooth (Ruiz et al., 2023) as the baseline, where the goal is to generate images of a target concept from text prompts using only a handful of reference images. Following the experimental setup of Ruiz et al. (2023), we evaluate performance using three standard metrics: PRES (lower is better), DINO similarity (higher is better), and CLIP-I similarity (higher is better). We consider DreamBooth baselines following the Imagen-based implementation, with and without the prior preservation loss (PPL),

while keeping all other hyperparameters identical. As shown in Table 7, WILD-Diffusion improves these metrics over DreamBooth under both settings (with and without PPL), which suggests that our method can enhance generation quality in large-scale text-conditioned diffusion models.

Table 7: Text-to-image results on the DreamBooth dataset. We highlight the best results in **bold**.

| Methods | PRES ↓ | DINO ↑ | CLIP-I ↑ |
|---|---|---|---|
| DreamBooth (w/ PPL) (Ruiz et al., 2023) | 0.493 | 0.684 | 0.815 |
| + WILD-Diffusion (Ours) | **0.478** | **0.696** | **0.823** |
| DreamBooth (w/o PPL) (Ruiz et al., 2023) | 0.664 | 0.712 | 0.828 |
| + WILD-Diffusion (Ours) | **0.643** | **0.715** | **0.830** |

## C.6 EXPERIMENTAL RESULTS OF THE ABLATION STUDY

To further investigate the role of the distributional divergence, we compare our method based on the Wasserstein distance with variants using KL-divergence, $\chi^2$-divergence, and $\alpha$-divergence. As illustrated in Figure 11, Wasserstein distance consistently outperforms the alternatives across different data sizes (20%, 50%, and 100%). Notably, the improvement seems most evident in the low-data regime, indicating that the Wasserstein distance may play a role in stabilizing training under limited data.

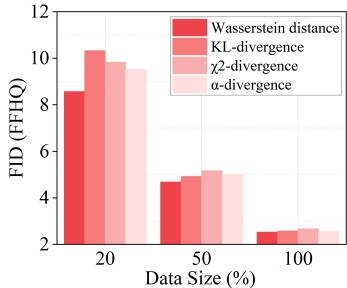

Figure 11: Ablation study on different distributional divergences for limited data generation on the FFHQ dataset. FID (lower is better) is reported under varying data sizes.

In addition, we compare our proposed WILD-Diffusion with commonly used augmentation techniques, including Mixup (Zhang et al., 2018), CutMix (Yun et al., 2019), and CutOut (DeVries & Taylor, 2017), based on the EDM-DDPM++ baseline (Karras et al., 2022). For these methods, we follow the default hyperparameters used in the original papers, which are also the standard configurations adopted in prior generative modeling work (Zhang et al., 2025). Specifically, we set the interpolation strength to $\alpha = 1$ for Mixup and CutMix, and use a $16 \times 16$ mask size for Cutout. The results are summarized in Table 8, which shows that WILD-Diffusion achieves the best performance among all compared approaches.

Table 8: Ablation study on data augmentation methods for FFHQ generation with $20\%$ training data. Results are reported in terms of FID using $50k$ samples.

| | |
|---|---|
| EDM-DDPM++ (Karras et al., 2022) | 10.02 |
| + WILD-Diffusion | **8.57** |
| + Mixup (Zhang et al., 2018) | 10.21 |
| + Cutmix (Yun et al., 2019) | 10.43 |
| + Cutout (DeVries & Taylor, 2017) | 10.25 |

We further analyze the computational efficiency of our method by measuring the relative running time under different data sizes (20%, 50%, and 100%). As summarized in Table 9, the training time

of the baseline EDM-DDPM++ (Karras et al., 2022) is normalized to "1". The results show that our method maintains almost identical running times across all data sizes, with values ranging from 1.20 to 1.22. This indicates that the performance gains of WILD-Diffusion come at negligible additional computational cost, thereby ensuring both effectiveness and efficiency.

Table 9: Training time analysis under varying data sizes (20%, 50%, and 100%). The training time of the baseline EDM-DDPM++ (Karras et al., 2022) is normalized to "1".

| Data size | Training time |
|-----------|---------------|
| 20% | 1.20 |
| 50% | 1.22 |
| 100% | 1.21 |

To provide a comprehensive analysis of computational overhead, we report wall-clock time, FLOPs, and peak GPU memory for both EDM-DDPM++ and WILD-Diffusion in Table 10. WILD-Diffusion increases training time by $1.21 \times$ and FLOPs by $1.25 \times$, while peak GPU memory increases by only 3%. These results indicate that our method adds minimal overhead relative to standard training.

Table 10: Comparison of training cost between EDM-DDPM++ (Karras et al., 2022) and WILD-Diffusion across wall-clock time, FLOPs, and peak GPU memory.

| Methods | Wall-clock time (h) | FLOPs (G) | GPU memory (GB) |
|---------|---------------------|-----------|-----------------|
| EDM-DDPM++ (Karras et al., 2022) | 26.4 | 137 | 16.32 |
| WILD-Diffusion (Ours) | 31.9 $(1.21\times)$ | 172 $(1.25\times)$ | 16.84 $(1.03\times)$ |

To further understand how the computational cost scales with the hyperparameters, we additionally examine the effects of the interval parameter $m$ and the number of inner ascent steps $K$. As shown in Figure 12, increasing $m$ reduces FLOPs rapidly and then stabilizes, since "Bi-level Interval Update" occur less frequently. In contrast, increasing $K$ leads to a "near-linear" growth in FLOPs due to additional forward–backward passes. These results highlight the controllable computational behavior of WILD-Diffusion.

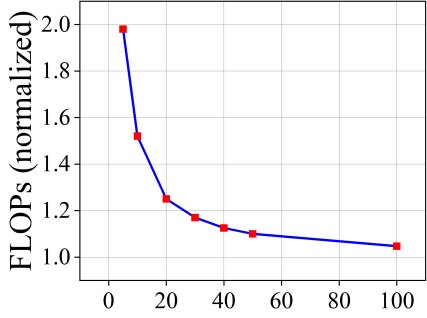

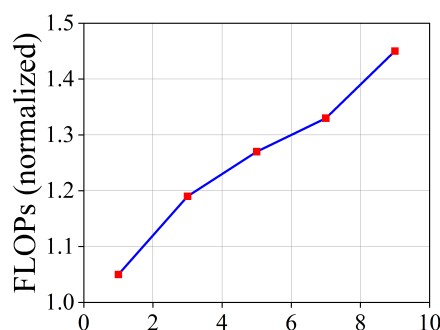

(a) Effect of the interval parameter $m$ on FLOPs.      (b) Effect of the number of inner steps $K$ on FLOPs.

Figure 12: Overall comparison of FLOPs under different support expansion configurations. (a) Relationship between FLOPs and interval parameter $m$. (b) Relationship between FLOPs and the number of inner steps $K$. The FLOPs of the baseline EDM-DDPM++ (Karras et al., 2022) is normalized to "1".

# D  USEFUL FACTS

In this section, we collect some facts used throughout the paper.

**Definition D.1** (*$f$-divergence*). *Let $f : (0, \infty) \to \mathbb{R}$ be a convex function with $f(1) = 0$. Let $P$ and $Q$ be two probability distributions on a measurable space $(\mathcal{X}, \mathcal{F})$. If $P \ll Q$ then the $f$-divergence is defined as*

$$\mathrm{D}_f(P\|Q) \triangleq \mathbb{E}_Q\left[f\left(\frac{\mathrm{d}P}{\mathrm{d}Q}\right)\right]$$

*where $\frac{\mathrm{d}P}{\mathrm{d}Q}$ is a Radon-Nikodym derivative and $f(0) \triangleq f(0+)$. Suppose that $Q(\mathrm{d}x) = q(x)\mu(\mathrm{d}x)$ and $P(\mathrm{d}x) = p(x)\mu(\mathrm{d}x)$ for some common dominating measure $\mu$, then we have*

$$\mathrm{D}_f(P\|Q) = \int q(x)f\left(\frac{p(x)}{q(x)}\right)\mathrm{d}\mu$$

The following are common $f$-divergences that used in this paper:

1. Kullback-Leibler (KL) divergence: $f(x) = x \log x$,

$$\mathrm{D}_{\mathrm{KL}}(P\|Q) \triangleq \int p(x) \log \frac{p(x)}{q(x)} \mu(\mathrm{d}x).$$

2. $\chi^2$-divergence: $f(x) = (x-1)^2$,

$$\chi^2(P\|Q) \triangleq \mathbb{E}_Q\left[\left(\frac{\mathrm{d}P}{\mathrm{d}Q} - 1\right)^2\right] = \int \frac{\mathrm{d}P^2}{\mathrm{d}Q} - 1$$

3. Total variation: $f(x) = \frac{1}{2}|x-1|$,

$$\mathrm{D}_{\mathrm{TV}}(P,Q) \triangleq \frac{1}{2}\mathbb{E}_Q\left[\left|\frac{\mathrm{d}P}{\mathrm{d}Q} - 1\right|\right] = \frac{1}{2}\int |\mathrm{d}P - \mathrm{d}Q|$$

4. $\alpha$-divergence (Wang et al., 2018a): $f(x) = \frac{x^\alpha - 1}{\alpha(\alpha-1)}$, $\alpha \in \mathbb{R} \setminus \{0,1\}$ and hence

$$\mathrm{D}_\alpha(P\|Q) = \frac{1}{\alpha(\alpha-1)}\mathbb{E}_Q\left[\left(\frac{\mathrm{d}P}{\mathrm{d}Q}\right)^\alpha - 1\right].$$

**Definition D.2** (*log-Sobolev inequality (Vempala & Wibisono, 2019)*). *Let $P$ be a probability measure with density $p$. We say that $p$ satisfies a log-Sobolev inequality with constant $C_{\mathrm{lS}}$ if, for any probability measure $q$,*

$$\mathrm{KL}(q \,\|\, p) \leq \frac{C_{\mathrm{lS}}}{2} \int \left\|\nabla \log \frac{q(x)}{p(x)}\right\|^2 q(x)\,\mathrm{d}x.$$

