# OpenReview forum: "WILD-Diffusion: A WDRO Inspired Training Method for Diffusion Models under Limited Data"
_ICLR.cc/2026/Conference — ICLR 2026 Poster_

### Official Review · Reviewer_rEDT · 2025-10-22

**Soundness:** 4
**Presentation:** 4
**Contribution:** 2
**Rating:** 4
**Confidence:** 4

**Summary:**

It is known that training diffusion models requires a lot of data. To mitigate this, the paper proposes a training method using Wasserstein Distributionally Robust Optimization (WDRO) called WILD-Diffusion. Using DRO, the paper can train the model to be more fitted on not only given sharp limited training data, but also near smooth distribution by expanding the support of the data. To do this, the paper uses the two-levels update strategy one for updating parameters and the other for updating samples. They also provide convergence analysis for the proposed method.

**Strengths:**

The paper is well written and provides rich theoretical analysis. Applying DRO to diffusion models for few-shot learning is a new idea and technically sounds. To prove the method works well, they show that the proposed method achieved high performance when the data is limited (e.g. 20%, 50% in Table 1). Also, it can be applied to existing diffusion models easily as they applied the method to various existing models such as EDM, Patch Diffusion and DeepCache.

**Weaknesses:**

- Applying DRO to diffusion models is already studied in previous works (e.g. Improved Diffusion-based Generative Model with Better Adversarial Robustness, ICLR 2025). While the main topics of them and this paper may different, the novelty of the proposed work may decrease.
- The datasets used in the experiments seem limited. These dataset have relatively narrow in topic, so perturbing them may effectively expand the dataset support. However, for larger and more complex dataset (e.g. ImageNet), expanding the support may introduce degradation by off-manifold samples.
- The comparisons with other few-shot diffusion model are only conducted on 100-shot and Animal Face datasets. In fact, since the numbers in Table 6 in the appendix show that other few-shot works also achieve comparable performance, it seems necessary to verify whether the performance gap remains consistent across a wider variety of datasets. Comparisons on other datasets such as CelebA, MNIST, and LSUN would strengthen the evaluation.

**Questions:**

- What are the key differences between this work and previous studies that applied DRO to diffusion models? (e.g. Improved Diffusion-based Generative Model with Better Adversarial Robustness)
- Is the proposed method robust even when the data manifold is highly complex?
- How much does the training time increase due to the two-level update strategy?
- Can the data generated by the proposed method but not belonging to the given data distribution be separately visualized?

---

> ### Author Response · Authors · 2025-11-21
>
> Thank you for your thoughtful comments and suggestions, and we try to address your questions and concerns below.
>
> **W1: Applying DRO to diffusion models is already studied in previous works ... While the main topics of them and this paper may different, the novelty of the proposed work may decrease.**
>
> **A:** We thank the reviewer for pointing out this related work. We agree that both papers are inspired by distributionally robust optimization (DRO), but they address different problems and employ substantially different DRO formulations and algorithms.
>
> **(1) Different problems and challenges**; The work [1] focuses on large-scale training when abundant data are available, and addresses the mismatch between the training and sampling distributions, i.e., exposure bias and the adversarial robustness issues of diffusion models. In contrast, our work is specifically motivated by the limited data setting, where the empirical data distribution is severely under-sampled and diffusion models suffer from overfitting (U-shaped FID curves, memorization of training samples; please see Figure 1 in our paper). Our goal is to overcome overfitting and improve generative quality under small datasets by expanding the support of the training distribution, rather than to improve robustness of intermediate states under standard data-rich training. To further clarify the difference, we also added a empirical comparison between Wang et al. (2025) [1] and our method under the limited data setting (Table 1 below). As shown, their method provides only marginal improvement over the EDM-DDPM++ baseline, whereas WILD-Diffusion yields substantially larger gains under 20% and 50% training data. This confirms that the method in Wang et al. can not address the overfitting issues arising in limited data diffusion training, which is the core problem tackled in our work.
>
> Table1: Comparison with Wang et al. (2025) [1] under limited data settings (20% and 50% of CIFAR-10).
>
> | $\text{Methods}$                 |    $20\\% $ data    |   $50\\%$ data    |
> | :------------------------------- | :----------------: | :--------------: |
> | $\text{EDM-DDPM++}$              |      $13.91$       |      $6.62$      |
> | $\text{+ Wang et al. (2025)}$    |      $13.63$       |      $6.49$      |
> | $\text{+ WILD-Diffusion (Ours)}$ | $12.14~(-12.72\\%)$ | $6.02~(-9.08\\%)$ |
>
> **(2) Different DRO formulations and algorithms;** The prior work develops a DRO around the intermediate distribution $q(x_{t+1})$ and shows that this is equivalent to adversarial training at each diffusion step. The adversarial examples are not stored and are only used as transient perturbations for robustness. In contrast, we formulate a **Wasserstein-ball DRO** around the **original data distribution** $p_{data}$, which is more suitable to model uncertainty and support expansion in the limited data setting. Using the dual formulation, we derive a surrogate loss and design a **bi-level interval update** procedure that explicitly optimizes and then reuses worst-case samples as an adaptive, theoretically-grounded data augmentation mechanism. To the best of our knowledge, WILD-Diffusion is the first work to apply Wasserstein DRO to expand the support of limited training data in diffusion models.
>
> In addition, We have made these distinctions clearer in the revised manuscript, in **Section 1.1** and **Related Work**.
>
> [1] Wang et al., "Improved Diffusion-based Generative Model with Better Adversarial Robustness." ICLR 2025

---

> ### Author Response · Authors · 2025-11-21
>
> **W2: The datasets used ...  more complex dataset (e.g. ImageNet), ... degradation by off-manifold samples.**
>
> **A:** We thank the reviewer for the helpful comment. We agree that the datasets used in our main experiments have relatively narrow coverage, and that evaluating the method on larger and more diverse datasets is important for assessing its generality.  Therefore, to directly address this concern, we have added experiments on the ImageNet dataset (Table 2), which is substantially larger and more complex.
>
> Table 2: FID results on the $20\\%$ ImageNet training dataset.
>
> | $\text{Methods}$                |   $\text{FID}$   |
> | :------------------------------ | :--------------: |
> | $\text{EDM-DDPM++}$             |      $4.57$      |
> | $\text{+WILD-Diffusion (Ours)}$ | $4.15~(-9.20\\%)$ |
>
> The results show that WILD-Diffusion also improves performance on ImageNet, confirming that support expansion does **not** introduce degradation from off-manifold samples even when the underlying dataset is large and complex. Together with our CIFAR-10, FFHQ, CelebA-HQ, and LSUN results, these findings suggest that WILD-Diffusion generalizes well across datasets of very different scales and complexity levels.
>
> **W3: The comparisons with other ... and LSUN would strengthen the evaluation.**
>
> **A:** We thank the reviewer for the helpful suggestion. The reviewer asked whether the performance gap between our method and prior few-shot baselines remains stable across more datasets. To address this concern, we added experiments on two additional datasets, CelebA-HQ and LSUN-Cat, following the reviewer’s recommendation. Across both datasets, our method achieves the lowest FID scores as shown in Table 3. The performance gap relative to prior few-shot methods is consistent with the original experiments. These results show that the improvements of WILD-Diffusion are not tied to a specific dataset and remain stable across diverse domains.
>
> Table 3: The FID results on few-shot generation with GAN architecture.
>
> | $\text{Methods}$     | $\text{CelebA-HQ} \to \text{FFHQ}$ | $\text{FFHQ} \to \text{LSUN-Cat}$ |
> | :------------------- | :--------------------------------: | :-------------------------------: |
> | $\text{DiffAugment}$ |              $11.20$               |              $20.18$              |
> | $\text{ADA}$         |              $10.08$               |              $19.34$              |
> | $\text{MAFP}$        |               $9.67$               |              $17.93$              |
> | $\text{Ours}$        |               $8.53$               |              $16.28$              |
>
>
> **Q1:  What are the ... to diffusion models?**
>
> **A:**  For a detailed answer, we kindly refer the reviewer to our response to **W1**, where we discuss this point thoroughly.
>
> **Q2: Is the proposed method robust even when the data manifold is highly complex?**
>
> **A:**  We appreciate the reviewer’s question on robustness under highly complex data manifolds. First, we have added experiments on **ImageNet** in the Table 2 in **W1**. ImageNet is substantially larger and more diverse than the datasets in our main experiments. On the 20% training data, WILD-Diffusion improves the FID of EDM-DDPM++ from $4.57$ to $4.15~(−9.20\\%)$, showing that our method still brings gains on a large-scale, highly complex dataset. Second, Section 4.2 and **Table 5** (*page 25*) evaluate WILD-Diffusion on the **LSUN-Church 256×256** dataset, which has high spatial resolution and complex scene structure. Across 20%, 50%, and 100% data settings, WILD-Diffusion consistently reduces FID over EDM-ADM (e.g., $7.74 \to 6.98$ at $20\\%$, $5.79 \to 5.13$ at $50\\%$).
>  Taken together, the results on ImageNet and high-resolution LSUN-Church indicate that WILD-Diffusion remains robust even when the underlying data manifold is large, high-dimensional, and visually complex.
>
> **Q3: How much does the training time increase due to the two-level update strategy?**
>
> **A:** We thank the reviewer for this question. As shown in **Table 9** of our submission, WILD-Diffusion increases training time by roughly **20%** compared with the EDM-DDPM++ baseline, which indicates that the extra operations (Bi-level Interval Update strategy) adds only a small overhead. To provide a complete analysis, we added **Table 10** in the revised manuscript. It reports wall-clock time, FLOPs, and GPU memory usage. WILD-Diffusion requires $1.21\times$ wall-clock time and $1.25\times$  FLOPs, while GPU memory usage increases by only $3\\%$. These results show that the computational overhead of our method remains small in practice.

---

> ### Author Response · Authors · 2025-11-21
>
> **Q4: Can the data generated by ... belonging to the given data distribution be separately visualized?**
>
> **A:** We thank the reviewer for this helpful suggestion. In the revised manuscript, we have added **Figure 5**, which separately visualizes the samples generated by WILD-Diffusion that do not belong to the given data distribution (i.e., the worst-case samples identified by our bi-level update).

---

> ### Author Response · Authors · 2025-11-26
>
> Dear Reviewer `rEDT`,
>
> Thank you for taking the time to review our paper and provide valuable feedback. As the discussion phase is nearing its conclusion, we would like to confirm if our responses from a few days ago have effectively addressed your concerns. If you have any additional comments, we will do our best to address them.
>
> Best regards,
>
> The authors of Submission **9149**.

---

> > ### Comment · Reviewer_rEDT · 2025-11-26
> >
> > Thank you for addressing the reviewer’s questions. Most of my concerns have been resolved, including the evaluation on complex data (ImageNet) and the comparison with the existing paper (Wang et al., 2025). The work is theoretically rich and demonstrates that DRO can be effectively applied in the limited-data setting, so the reviewer would like to change the recommendation to accept.
> >
> > However, although the direction is different, there have been related approaches in prior work, and the proposed method is particularly effective only under the restricted scenario of limited domains and limited data. Since recently large-scale data is abundant, the practical usefulness of such a setting may be somewhat limited. Therefore, the reviewer would like to revise my score to marginally accept.

---

> > > ### Author Response · Authors · 2025-11-27
> > >
> > > We thank the reviewer again for the helpful comments. We agree that large-scale data have become increasingly available in many benchmarks. However, in many real world scenarios, collecting and cleaning such large-scale datasets can be expensive, time consuming, and sometimes infeasible, which significantly constrains the deployment of generative models in practice [1-2]. Our work specifically targets this limited data setting and provides a principled way to improve generation quality and mitigate overfitting when only a small number of samples are available.
> > >
> > > [1] Zhang et al., "Training diffusion-based generative models with limited data." ICML 2025
> > >
> > > [2] Karras et al., "Training Generative Adversarial Networks with Limited Data."  NeurIPS 2020

---

### Official Review · Reviewer_GEiA · 2025-11-01

**Soundness:** 3
**Presentation:** 2
**Contribution:** 3
**Rating:** 6
**Confidence:** 3

**Summary:**

This paper addresses overfitting when training diffusion models with limited data, observing U-shaped FID curves and reduced diversity as training proceeds on small subsets.
Concretely, a training framework inspired by Wasserstein Distributionally Robust Optimization (WDRO) is proposed, termed WILD-Diffusion,  which expands the support of the training distribution by iteratively generating worst-case samples within a Wasserstein ball around the empirical data and mixing them into training.
To keep this tractable, the WDRO inner max is dualized into a surrogate loss and optimized with a bi-level interval update (periodically refreshing adversarial samples, then standard parameter updates), and the paper proves a convergence guarantee tailored to the WDRO+diffusion setting.
Experiments on CIFAR-10, FFHQ, CelebA-HQ, LSUN-Church and several backbones report >10% FID reductions with only 20% data and SoTA FID with as few as 100 images, both with and without pretraining.

**Strengths:**

This paper investigates an interesting and important problem for the research community. The proposed approach is technically sound, viewing data scarcity through a distributionally robust lens and converting the push for more data into adversarial, Wasserstein-bounded augmentations that expand support without drifting off-distribution. The optimization is practical: a tractable surrogate objective with a simple bi-level refresh of worst-case samples that plugs into standard training loops and works with or without pretraining. Theoretical analysis supports the design, and experiments show meaningful FID gains in very low–data regimes, demonstrating utility beyond incremental tweaks. Overall, the method is conceptually clean, easy to adopt, and delivers strong sample-efficiency improvements in settings where diffusion models typically overfit.

**Weaknesses:**

1. While the dual surrogate makes the WDRO objective tractable, it can mis-specify the true worst-case set and depends on strong-duality assumptions.

2. The convergence proof requires Lipschitz/log-Sobolev–style assumptions for data and score error bounds; these may not hold for real high-res image manifolds.

3. The bi-level update generates **worst-case** samples and mixes them with real data, but the paper doesn’t quantify how often those samples are usefully hard versus toxic.

**Questions:**

1. How is the robustness radius selected across datasets and data regimes?

2. How closely does the dual surrogate approximate the true worst-case set?

3. What are the wall-clock/GPU-hour and peak-memory overheads of the adversarial refresh loop relative to standard training?

4. Do the improvements extend to high-resolution generation and text-to-image settings, and what changes (if any) are needed to the loss, model, or optimization?

---

> ### Author Response · Authors · 2025-11-21
>
> Thank you for your thoughtful comments and suggestions, and we try to address your questions and concerns below.
>
> **W1: While the dual surrogate makes the WDRO objective tractable, it can mis-specify the true worst-case set and depends on strong-duality assumptions.**
>
> **A:** We appreciate the reviewer’s careful remark. First, the mapping $p\mapsto W_{c}(p, q)$ is convex in the space of probability measures. As taking $p=q$ yields $W_{c}(q,q)=0$​​, Slater’s condition [1] holds and we may apply standard (infinite dimensional) duality results [2, 3] to obtain
>
> $$
> \sup_{p:W_{c}(p, q)}\int f(x)dp(x) =\sup_{p:W_{c}(p, q)} \inf_{\gamma>0} \\{\int f(x)dp(x)-\gamma W_{c}(p,q)+\gamma \rho\\}
> = \inf_{\gamma>0} \sup_{p:W_{c}(p, q)} \\{\int f(x)dp(x)-\gamma W_{c}(p,q)+\gamma \rho \\}.
> $$
>
> Therefore,  the dual surrogate is equivalent to the original WDRO objective and thus does not introduce mis-specification of the worst-case set.
>
> [1] Slater, M., "Lagrange multipliers revisited." In Traces and emergence of nonlinear programming 2013.
>
> [2] Luenberger, D. G., "Optimization by vector space methods."  John Wiley & Sons 1997.
>
> [3] Sinha et al.,  "Certifying Some Distributional Robustness with Principled Adversarial Training." ICLR 2018
>
> **W2: The convergence proof requires Lipschitz/log-Sobolev–style assumptions for data and score error bounds; these may not hold for real high-res image manifolds.**
>
> **A:** Thanks for raising this question. Indeed, Lipschitz continuity and log-Sobolev conditions are standard assumptions in the theoretical analysis of score-based generative models [1, 2]. Similar assumptions are commonly adopted in prior convergence studies to obtain tractable bounds. These assumptions serve as sufficient (but not necessary) regularity conditions for the theoretical analysis. In our work, the empirical results on real high-resolution datasets (e.g., LSUN-Church-256×256 in **Table 5** in Appendix C.3) demonstrate stable optimization and consistent performance improvements, indicating that our method performs well even beyond the idealized theoretical setting.
>
> [1] Lee et al., "Convergence for score-based generative modeling with polynomial complexity." NeurIPS 2022
>
> [2] Block et al., "Generative modeling with denoising auto-encoders and langevin sampling." arXiv 2020
>
> **W3: The bi-level update generates worst-case samples and mixes them with real data, but the paper doesn’t quantify how often those samples are usefully hard versus toxic.**
>
> **A:** We thank the reviewer for raising this important point. The effectiveness of worst-case samples indeed depends on whether they provide usefully hard training signals rather than introducing toxic. We address this concern from both an empirical and methodological perspective.
>
> **(1)** From the methodological perspective: Our "Bi-level Interval Update" strategy is designed to produce **bounded perturbations** around the data manifold, rather than arbitrary off-distribution samples (such as Mixup, Cutmix, and Cutout in Table 8 in the paper). Specifically, The inner maximization is regularized by the Wasserstein-ball constraint, which ensures that the worst-case samples remain **close to the real data distribution** (please see Equation 11). This mechanism inherently prevents the update from drifting too far from the data manifold and keeps the synthesized samples "hard but meaningful".
>
> **(2)** From the empirical perspective: To make this clearer, we have added an empirical analysis of the worst-case samples in Table 1 (below). We report their FID relative to the original dataset and the training stability metrics (gradient variance [1]). The results shows that these worst-case samples remain **near-distribution** (small FID score) and no toxic behavior (i.e., no toxic gradients) are observed.
>
> Table 1: Characterization of worst-case samples generated by WILD-Diffusion on CIFAR-10 datasets. The gradient variance of the real images is normalized to "1".
>
> | $\text{Metric}$                           | $\text{Worst-case Images}$ |
> | :---------------------------------------- | :------------------------: |
> | $\text{FID} \downarrow$                   |          $0.2697$          |
> | $\text{Gradient Variance [1]} \downarrow$ |          $< 1.05$          |
>
> [1] Ghosh et al., "How gradient estimator variance and bias impact learning in neural networks." ICLR 2023

---

> ### Author Response · Authors · 2025-11-21
>
> **Q1: How is the robustness radius selected across datasets and data regimes?**
>
> **A:** In our method (Algorithm 1), the robustness radius $\rho$ in the WDRO formulation is absorbed into the dual weight $\gamma$ (since $\gamma \propto 1/\rho$). Thus, selecting $\gamma$ corresponds directly to controlling the size of the Wasserstein uncertainty set. Across different datasets and data regimes, we set $\gamma=1$ following the same principle used in prior WDRO literature [1-2]. To validate this choice in our problem, we evaluate several $\gamma$ values and observe that $\gamma=1$ achieves the best performance on both the $50\\%$ FFHQ dataset (as shown in Section 4.1 and Figure 4c of the manuscript) and $20\\%$ CIFAR-10 dataset. This supports using the same $\gamma=1$ uniformly across datasets without dataset-specific tuning.
>
> Table 2: Impact of the dual weight $\gamma$ (equivalently, robustness radius $\rho$) on model performance.
>
> | $\gamma$                 | $0.001$ | $0.01$  |  $0.1$  |  $ 1$   |  $10$   |  $100$  |
> | :----------------------- | :-----: | :-----: | :-----: | :-----: | :-----: | :-----: |
> | $\text{FFHQ}~(50\\%)$     | $5.17$  | $4.76$  | $4.69$  | $4.68$  | $4.95$  | $5.03$  |
> | $\text{CIFAR-10}~(20\\%)$ | $14.33$ | $13.98$ | $12.27$ | $12.14$ | $12.44$ | $13.66$ |
>
> [1] Huang et al., "An Effective Manifold-based Optimization Method for Distributionally Robust Classification." ICLR 2025
>
> [2] Volpi et al., "Generalizing to Unseen Domains via Adversarial Data Augmentation." NeurIPS 2018
>
> **Q2: How closely does the dual surrogate approximate the true worst-case set?**
>
> **A:**  As discussed in **W1**, since the Slater condition hold in our WDRO formulation, the dual problem is exactly equivalent to the primal WDRO objective. Therefore, the dual surrogate does not approximate but coincides with the true worst-case value, and therefore does not mis-specify the worst-case set.
>
> **Q3: What are the wall-clock/GPU-hour and peak-memory overheads of the adversarial refresh loop relative to standard training?**
>
> **A:** We thank the reviewer for this question. We have measured the overhead of the adversarial refresh loop on our main CIFAR-10 experiment (EDM-DDPM++ baseline). The results are reported in Table 3 (**Table 10** of the revised manuscript).  Standard EDM-DDPM++ training takes  $26.4$ hours. With WILD-Diffusion and the adversarial refresh loop enabled, the total training time increases to $31.9$ hours, i.e., a $1.21\times$ wall-clock (and GPU-hour) overhead under the same hardware and batch size. The peak memory usage of the baseline is $16.32 \text{GB}$, while WILD-Diffusion uses $16.84 \text{GB}$, corresponding to only a $3\\%$ increase  in peak memory.
>
> Table 3: Training cost and generative performance comparison between EDM-DDPM++ and WILD-Diffusion in terms of wall-clock time, FLOPs, GPU memory usage, and FID on the 20% CIFAR-10 ($32\times32$) dataset.
>
> | $\text{Methods}$                | $\text{Wall-clock time (h)}$ | $\text{FLOPs (G)}$ | $\text{GPU memory (GB)}$ |    $\text{FID}$    |
> | :------------------------------ | :--------------------------: | :----------------: | :----------------------: | :----------------: |
> | $\text{EDM-DDPM++}$             |            $26.4$            |       $137$        |         $16.32$          |      $13.91$       |
> | $\text{+WILD-Diffusion (Ours)}$ |     $31.9~(1.21\times)$      | $172~(1.25\times)$ |   $16.84~(1.03\times)$   | $12.14~(-12.72\\%)$ |

---

> ### Author Response · Authors · 2025-11-21
>
> **Q4: Do the improvements extend to high-resolution generation and text-to-image settings, and what changes (if any) are needed to the loss, model, or optimization?**
>
> **A:** We appreciate the reviewer’s thoughtful question. Our method is model-agnostic and architecture-free: WILD-Diffusion operates entirely at the data distribution level, so it applies to high-resolution generation and text-conditioned diffusion models **without** modifying the model and optimization procedure.
>
> **High-resolution settings.** Our original manuscript already includes high-resolution experiments on LSUN-Church-256×256 (Section 4.2; Appendix C.3, Table 5), where we observe consistent gains under limited data settings. These results suggest that WILD-Diffusion can scale to higher-resolution generative tasks.
>
> **Text-to-image settings.** To directly address the reviewer’s concern, we added a DreamBooth [1] few-shot personalization experiment. As reported in Table 4, adding WILD-Diffusion to the DreamBooth pipeline reduces overfitting (lower PRES) and improves subject fidelity (higher DINO and CLIP-I) with only 3–5 training images. These results further suggest that our plug-and-play framework benefit large-scale text-conditioned diffusion models.
>
> Table 4: Text-to-image results on the DreamBooth dataset.
>
> | $\text{Methods}$                  | $\text{PRES} \downarrow$ | $\text{DINO} \uparrow$ | $\text{CLIP-I} \uparrow$ |
> | --------------------------------- | :----------------------: | :--------------------: | :----------------------: |
> | $\text{DreamBooth (Imagen)}$      |         $0.493$          |        $0.684$         |         $0.815$          |
> | $\text{+WILD-Diffusion (Imagen)}$ |         $0.478$          |        $0.696$         |         $0.823$          |
>
> [1] Ruiz et al., "DreamBooth: Fine Tuning Text-to-Image Diffusion Models for Subject-Driven." CVPR 2023

---

> ### Author Response · Authors · 2025-11-26
>
> Dear Reviewer `GEiA`,
>
> Thank you for taking the time to review our paper and provide valuable feedback. As the discussion phase is nearing its conclusion, we would like to confirm if our responses from a few days ago have effectively addressed your concerns. If you have any additional comments, we will do our best to address them.
>
> Best regards,
>
> The authors of Submission **9149**.

---

### Official Review · Reviewer_hDug · 2025-11-01

**Soundness:** 3
**Presentation:** 3
**Contribution:** 2
**Rating:** 6
**Confidence:** 4

**Summary:**

The paper proposes WILD‑Diffusion, a WDRO‑inspired, plug‑and‑play training framework for diffusion models under limited data. It extends Wasserstein Distributionally Robust Optimization to train diffusion models. Furthermore, the authors provide a convergence guarantee despite the coupling between the diffusion process and WDRO. Empirically, WILD‑Diffusion shows competitive performance in both fine-tuning and from-scratch setting.

**Strengths:**

1. Theory‑grounded, practical framework with broad applicability.
- Beyond heuristic data augmentation, the approach is anchored in WDRO and comes with a convergence guarantee. It shows strong performance in limited‑data regimes in both from‑scratch training and fine‑tuning scenarios, accompanied by ablations that indicate robust gains.

**Weaknesses:**

1. Clarification of few‑shot comparisons and OOD adaptation.
- In Section 4.3 (Experiments on Few‑shot Generation), Table 2 compares fine‑tuning settings with pretraining, but the original pretrained model’s performance is not shown alongside. Moreover, the FFHQ‑pretrained fine‑tuning appears in‑distribution for faces; it would be informative to evaluate out‑of‑distribution adaptation, for example transferring to a stylized or non‑face domain such as a Pokémon dataset, to assess robustness under distribution shift.

**Questions:**

1. Overhead of bi‑level interval updates.
- How does the proposed bi‑level interval update compare to standard full fine‑tuning in terms of wall‑clock time, training steps, and memory/compute overhead? A quantitative breakdown would help practitioners assess the cost–benefit trade‑off.

2. Comparison to attention‑only fine‑tuning under limited data.
- Prior work on fine‑tuning diffusion models with limited data [1] suggests that attention‑only fine‑tuning can prevent overfitting while remaining competitive. Has WILD‑Diffusion been compared against such parameter‑efficient baselines, and if so, under identical data budgets and evaluation protocols?

3. Large scale model experiments on limited data
- The proposed method demonstrates strong effectiveness under limited-data conditions, but its impact could be further amplified by including experiments on large-scale text-to-image diffusion models. Showing that WILD-Diffusion maintains or improves few-shot performance in such settings would better highlight its generality and practical relevance.

References

[1] Moon et al., Fine-tuning diffusion models with limited data

---

> ### Author Response · Authors · 2025-11-21
>
> Thank you for your thoughtful comments and suggestions, and we try to address your questions and concerns below.
>
> **W1: Clarification of few‑shot comparisons and OOD adaptation. In Section 4.3 ... robustness under distribution shift.**
>
> **A:** We thank the reviewer for the helpful comments.
>
> **(1)** We appreciate the reviewer for pointing out this omission. In the revised manuscript, we have added the performance of the original pretrained models. The updated table (shown below as Table 1) now reports the baseline FID scores of the pretrained StyleGAN-v2 and EDM-NCSN++ models across different datasets.
>
> Table 1: Baseline performance  $(\text{FID}\downarrow)$  of the pretrained StyleGAN-v2 and EDM-NCSN++ models.
>
> | $\text{Methods}$     | $\text{Obama}$ | $\text{Grumpy}$ | $\text{Panda}$ | $\text{Cat}$ | $\text{Dog}$ |
> | :------------------- | :------------: | :-------------: | :------------: | :----------: | :----------: |
> | $\text{StyleGAN-v2}$ |    $80.20$     |     $48.90$     |    $34.27$     |   $71.71$    |   $130.19$   |
> | $\text{EDM-NCSN++}$  |    $37.10$     |     $29.94$     |    $10.81$     |   $36.88$    |   $57.14$    |
>
> **(2)** Following the reviewer’s suggestion, we have additionally evaluated OOD adaptation by transferring pretrained models from FFHQ (face domain) to a stylized, non-face domain (Pokémon-64$\times$64). This experiment aims to examine the robustness of the models under substantial distribution shifts. As shown in Table 2, incorporating WILD-Diffusion consistently improves generation performance across both OOD settings.
>
> Table 2: Results $(\text{FID}\downarrow)$ of few-shot generation performance under significant distribution shifts.
>
> | $\text{Methods}$          | $\text{FFHQ}\to\text{Pokemon}$ |
> | :------------------------ | :----------------------------: |
> | $\text{EDM-NCSN++}$       |            $25.89$             |
> | $+ \text{WILD-Diffusion}$ |            $21.32$             |
>
> **Q1: Overhead of bi‑level interval updates. How does the proposed ... practitioners assess the cost–benefit trade‑off.**
>
> **A:** We thank the reviewer for raising this practical question. To provide a clearer analysis of the computational overhead, we added **Table 10** in the revised manuscript. The table reports wall-clock time, FLOPs, and GPU memory usage for both EDM-DDPM++ (baseline) and our WILD-Diffusion. The results show that WILD-Diffusion requires $1.21\times$ the training time and $1.25\times$ the FLOPs compared to standard training. GPU memory usage increases by only 3%, which indicates that the method introduces minimal additional computational cost. In addition, the total number of parameter-update steps remains the same as in standard training. WILD-Diffusion adds only a lightweight distribution-level update, and this update contributes negligible overhead relative to the main optimization loop.
>
> Table 3: Training cost and generative performance comparison between EDM-DDPM++ and WILD-Diffusion in terms of wall-clock time, FLOPs, GPU memory usage, and FID on the 20% CIFAR-10 ($32\times32$) dataset.
>
> | $\text{Methods}$                | $\text{Wall-clock time (h)}$ | $\text{FLOPs (G)}$ | $\text{GPU memory (GB)}$ |    $\text{FID}$    |
> | :------------------------------ | :--------------------------: | :----------------: | :----------------------: | :----------------: |
> | $\text{EDM-DDPM++}$             |            $26.4$            |       $137$        |         $16.32$          |      $13.91$       |
> | $\text{+WILD-Diffusion (Ours)}$ |     $31.9~(1.21\times)$      | $172~(1.25\times)$ |   $16.84~(1.03\times)$   | 12.14 (-12.72%) |

---

> ### Author Response · Authors · 2025-11-21
>
> **Q2: Comparison to attention‑only fine‑tuning under limited data. ... evaluation protocols?**
>
> **A:** We thank the reviewer for the helpful suggestion. We would like to clarify that $\text{A}^3\text{FT}$  [1] and WILD-Diffusion address **orthogonal** aspects (Model parameter vs. Data distribution) of the limited data diffusion generation problem. $\text{A}^3\text{FT}$ reduces the number of trainable parameters by restricting updates to a small subset of the network, which improves parameter efficiency and reduces overfitting. In contrast, WILD-Diffusion modifies the **data distribution** rather than the model parameters. It uses a Wasserstein Distributionally Robust Optimization (WDRO) formulation to create additional training samples inside a Wasserstein uncertainty set centered at the limited data distribution. These two approaches therefore operate at **different aspects** of the training pipeline, and their objectives are not identical. As a result, a direct comparison may not fully capture the complementary nature of the two methods.
>
> Nevertheless, WILD-Diffusion is inherently **plug-and-play** and can be combined with $\text{A}^3\text{FT}$. To address the reviewer’s question, **we added a new experiment** in the revised manuscript that integrates WILD-Diffusion with $\text{A}^3\text{FT}$. The results in **Table 4** show that WILD-Diffusion improves the generation quality of $\text{A}^3\text{FT}$, which suggests that our method is compatible with parameter-efficient tuning approaches and provides complementary benefits.
>
> Table 4: Results $(\text{cFID}\downarrow)$ of few-shot generation performance under significant distribution shifts. The **experimental setup** follows that of $\text{A}^3\text{FT}$.
>
> | $\text{Methods}$          | $\text{ImageNet}\to\text{Pokemon}$ | $\text{FFHQ}\to\text{MetFaces}$ |
> | :------------------------ | :--------------------------------: | :-----------------------------: |
> | $\text{A}^3\text{FT}$     |              $38.40$               |             $30.99$             |
> | $+ \text{WILD-Diffusion}$ |              $34.67$               |             $27.33$             |
>
> **Q3: Large scale model experiments on limited data. The proposed ... its generality and practical relevance.**
>
> **A:** We thank the reviewer for the suggestion. We have **added an experiment** on large-scale text-to-image diffusion models in the revised manuscript. Specifically, we integrate WILD-Diffusion into a DreamBooth [1] method (trained on Imagen), which represents a standard large-scale text-to-image setting. As shown in **Table 5,** WILD-Diffusion reduces overfitting (lower PRES) and improves subject fidelity (higher DINO and CLIP-I) under limited data. This suggests that our framework can scale to large text-conditioned models without requiring architectural changes.
>
> Table 5: Text-to-image results on the DreamBooth dataset.
>
> | $\text{Methods}$                  | $\text{PRES} \downarrow$ | $\text{DINO} \uparrow$ | $\text{CLIP-I} \uparrow$ |
> | --------------------------------- | :----------------------: | :--------------------: | :----------------------: |
> | $\text{DreamBooth [1] (Imagen)}$  |         $0.493$          |        $0.684$         |         $0.815$          |
> | $\text{+WILD-Diffusion (Imagen)}$ |         $0.478$          |        $0.696$         |         $0.823$          |
>
> [1] Ruiz et al., "DreamBooth: Fine Tuning Text-to-Image Diffusion Models for Subject-Driven." CVPR 2023

---

> ### Author Response · Authors · 2025-11-26
>
> Dear Reviewer `hDug`,
>
> Thank you for taking the time to review our paper and provide valuable feedback. As the discussion phase is nearing its conclusion, we would like to confirm if our responses from a few days ago have effectively addressed your concerns. If you have any additional comments, we will do our best to address them.
>
> Best regards,
>
> The authors of Submission **9149**.

---

> ### Comment · Reviewer_hDug · 2025-11-26
>
> Thank you for the substantial and thoughtful revisions. The new experiments addressing my earlier concerns, including OOD adaptation, computational overhead, and compatibility with parameter-efficient tuning, are much appreciated.
>
> I have one remaining clarification regarding the DreamBooth experiment in Table 5. The reported PRES, DINO, and CLIP-I values appear similar to those presented in Figure 6 of the original DreamBooth paper, but the manuscript does not specify the exact setting under which the baseline numbers were obtained. To help readers interpret the comparison more precisely, could the authors briefly describe the specific DreamBooth configuration used for the baseline?
>
> If the baseline indeed corresponds to Figure 6, that figure includes both the standard setting and the variant without prior preservation. Since prior preservation can significantly affect PRES and fidelity, including the w/o prior preservation baseline as well would offer a more complete and fair comparison.
>
> Overall, the manuscript has improved considerably, and I appreciate the authors’ efforts in the revision. I look forward to the clarification.

---

> > ### Author Response · Authors · 2025-11-27
> >
> > We thank the reviewer for the helpful clarification comment. In Table 5, the baseline uses the Imagen-based DreamBooth setting **with the prior-preservation loss (w/ PPL)** enabled. All other experimental settings follow Ruiz et al. (CVPR 2023). We now state this configuration explicitly in the revised manuscript, in both Appendix **C.5** and Table **7**.
> >
> > To provide a more complete and fair comparison, we added an additional experiment in which DreamBooth is trained **without the prior preservation loss (w/o PPL)**, which we obtain by disabling the prior preservation loss while keeping all other hyper-parameters identical. The new results are included in the revised Table 7. We observe that our method (+WILD-Diffusion) still achieves lower PRES and higher DINO/CLIP-I than both DreamBooth variants, so the main conclusions of the paper remain unchanged.
> >
> > | $\text{Methods}$                                 | $\text{PRES} \downarrow$ | $\text{DINO} \uparrow$ | $\text{CLIP-I} \uparrow$ |
> > | ------------------------------------------------ | :----------------------: | :--------------------: | :----------------------: |
> > | $\text{DreamBooth (Imagen)}~\text{w/ PPL}$       |         $0.493$          |        $0.684$         |         $0.815$          |
> > | $\text{+WILD-Diffusion (Imagen)}~\text{w/ PPL}$  |         $0.478$          |        $0.696$         |         $0.823$          |
> > | $\text{DreamBooth (Imagen)}~\text{w/o PPL}$      |         $ 0.664$         |        $0.712$         |         $0.828$          |
> > | $\text{+WILD-Diffusion (Imagen)}~\text{w/o PPL}$ |         $0.617$          |        $0.715$         |         $0.830$          |

---

> ### Comment · Reviewer_hDug · 2025-11-27
>
> Thank you for the detailed clarification. With the added explanations and experiments, my questions are fully addressed. I am satisfied with the revision and will keep my original positive evaluation for accepting this paper.

---

> > ### Author Response · Authors · 2025-11-28
> >
> > We thank the reviewer again for the helpful comments.

---

### Official Review · Reviewer_vXtY · 2025-11-01

**Soundness:** 2
**Presentation:** 2
**Contribution:** 2
**Rating:** 4
**Confidence:** 2

**Summary:**

The paper proposes WILD-Diffusion, a novel framework that adapts Wasserstein Distributionally Robust Optimization (WDRO) to diffusion models to promote robust training under limited-data regimes via  constructing worst-case distributions that are close to the limited data distribution which effectively expands the support of the training data and mitigates overfitting. Experiments on various benchmark datasets show consistent performance improvements, especially when data are scarce (e.g., N=100). Unlike traditional adversarial training, WILD-Diffusion is a plug-and-play procedure that requires no architectural changes and can be used with or without pretraining.

**Strengths:**

- The paper tackles a timely and relevant problem—improving diffusion model performance with limited data—supported by both theoretical grounding and extensive experiments.
- The overall presentation is clear, and the method is well motivated with sound empirical results.

**Weaknesses:**

- **Incomplete reporting of computational overhead** The paper reports a *relative* training-time increase of 1.20–1.22× compared to the baseline (Table 8), but does not provide absolute wall-clock time, FLOPs, or GPU memory usage. Since the method targets data-scarce scenarios where efficiency is important, providing these absolute cost measurements would significantly strengthen the practical relevance. Additionally, the paper should include scaling trends with respect to the inner ascent steps *K* and interval *m*. Table 8 also shows minimal overhead variation across dataset sizes—please clarify what the dominant bottleneck is and which component contributes most to runtime.
- **Mismatch between baselines and problem focus** The paper frames the core contribution as a *data-efficient training strategy for diffusion models*, yet some baselines are orthogonal to this goal. For instance, DeepCache (Table 1) focuses on *inference-time acceleration*, not robustness to limited data, and Patch Diffusion primarily aims to *reduce architectural training cost*, rather than address data scarcity. It would help to clarify whether these methods are included to demonstrate orthogonality or to claim direct superiority. In the current form, these comparisons blur the paper’s main message.
- **Insufficiently controlled augmentation baselines** In Table 7, WILD-Diffusion is compared with basic augmentation methods such as Mixup and CutMix, which perform even worse than the vanilla model. However, there is no indication that these techniques were tuned (e.g., optimal mixing ratios, label smoothing, class-aware mixing, or dataset-specific configurations). Without proper tuning and control, these results may underestimate augmentation baselines and overstate WILD’s relative advantage.
- **Extension to other domain tasks** Experiments are primarily conducted on low-resolution unconditional image generation. To better support claims of broad applicability, the method should also be validated on more practical and diverse settings such as class-conditional generation or text-to-image personalization (e.g., DreamBooth-style few-shot fine-tuning). Evaluating WILD-Diffusion on pretrained text-to-image models under data-scarce adaptation would demonstrate stronger real-world relevance.

**Questions:**

Refer to the above.

---

> ### Author Response · Authors · 2025-11-21
>
> Thank you for your thoughtful comments and suggestions, and we try to address your questions and concerns below.
>
> **W1:  Incomplete reporting of computational overhead. The paper reports a ... which component contributes most to runtime.**
>
> **A:** Thank you for pointing out the need for a more comprehensive report on computational overhead.
>
> **(1)** We agree that absolute measurements are important for assessing the practical efficiency of our method. In response, we have added **Table 10** to the revised manuscript, which reports the absolute wall-clock time, FLOPs (average), and GPU memory usage for both EDM-DDPM++ (baseline) and our WILD-Diffusion.
>
> Table 1: Training cost and generative performance comparison between EDM-DDPM++ and WILD-Diffusion in terms of wall-clock time, FLOPs, GPU memory usage, and FID on the 20% CIFAR-10 ($32\times32$) dataset.
>
> | $\text{Methods}$                | $\text{Wall-clock time (h)}$ | $\text{FLOPs (G)}$ | $\text{GPU memory (GB)}$ |    $\text{FID}$    |
> | :------------------------------ | :--------------------------: | :----------------: | :----------------------: | :----------------: |
> | $\text{EDM-DDPM++}$             |            $26.4$            |       $137$        |         $16.32$          |      $13.91$       |
> | $\text{+WILD-Diffusion (Ours)}$ |     $31.9~(1.21\times)$      | $172~(1.25\times)$ |   $16.84~(1.03\times)$   | 12.14 (-12.72%) |
>
> As shown in Table 1, our method increases the training wall-clock time by only $1.21\times$, and the FLOPs by  $1.25\times$ compared to baseline EDM-DDPM++. The GPU memory usage increases only marginally by about 3%, which confirms that our method introduces minimal additional memory overhead.  Hence, these results demonstrate that WILD-Diffusion improves generative performance in limited data settings while incurring only a small increase in training cost, thereby preserving its practical applicability.
>
> **(2)** We thank the reviewer for highlighting the importance of reporting the scaling behavior with respect to the interval parameter $m$ and the number of steps $K$. In the revised manuscript, we have added **Figure 12**, which provides a detailed analysis of how FLOPs vary under different parameter settings. **Figure 12(a)** shows that increasing the interval $m$ substantially reduces FLOPs, as the updates are invoked less frequently. The computational cost decreases rapidly and then stabilizes as $m$ grows. **Figure 12(b)** illustrates the scaling trend with respect to the number of steps $K$, where FLOPs increase in an **near-linear** manner because each step introduces an additional forward–backward pass during the bi-level interval update. Together, these results clearly characterize the computational behavior of WILD-Diffusion with respect to both hyperparameters $m$ and $K$, and help clarify the trade-off between computational overhead and performance (see Figure 3).
>
> **(3)** The overhead remains similar across different dataset sizes because the dominant bottleneck is the additional "forward–backward" passes introduced by the **Bi-level Interval Update**, which are independent of the dataset size. Data loading and other dataset-dependent components contribute negligibly to the overall runtime. Consequently, the runtime increase is primarily determined by the extra iterative computations controlled by $K$ and $m$, which is consistent with our scaling analysis in Figure 12.
>
> **W2: Mismatch between baselines and problem focus. The paper frames ... the paper’s main message.**
>
> **A:** We thank the reviewer for raising this concern. We include DeepCache [1] and Patch Diffusion [2] to show that WILD-Diffusion is a **plug-and-play training framework** that can be added to different diffusion methods, even if they focus on acceleration or architectural changes rather than limited data training. These methods are **not** direct baselines for our setting, but including them shows that our approach can work alongside these techniques. We have clarified this point in the revised manuscript (Section 4.2).
>
> [1] Ma et al., "Deepcache: Accelerating diffusion models for free." CVPR 2024
>
> [2] Wang et al., "Patch diffusion: Faster and more data-efficient training of diffusion models." NeurIPS, 2023

---

> ### Author Response · Authors · 2025-11-21
>
> **W3: Insufficiently controlled augmentation baselines. In Table 7, WILD-Diffusion is ...  may underestimate augmentation baselines and overstate WILD’s relative advantage.**
>
> **A:** We appreciate the reviewer’s concern regarding augmentation baselines. For Mixup [1], CutMix [2], and Cutout [3], we follow the default hyperparameters used in the original papers, which are also the standard configurations adopted in prior generative modeling work [4]. Specifically, we set the hyperparameter $\alpha=1$ (which controls the strength of interpolation between feature-target pairs for both Mixup and CutMix) and use a Cutout mask size of $16\times16$ pixels, consistent with standard practice for low-resolution datasets.
>
> To further address the reviewer’s concern,  we conducted a small-scale sensitivity study by varying $\alpha \in$ {0.2, 0.5, 1} and mask sizes {8$\times$8, 12$\times$12, 16$\times$16\}.  The results are shown in Table 2. Across all  hyperparameter settings, the augmentation baselines consistently remain below the performance of WILD-Diffusion (e.g., $\text{FID} = 8.57$ ). This indicates that the default configurations already reflect their behavior under the limited data setting, and that the performance gap is **not** due to under-tuning.
>
> Table 2: Small-scale hyperparameter sensitivity analysis for augmentation baselines.
>
> | $\text{Methods}$    | $ \alpha=0.2$ | $ \alpha=0.5$ | $\alpha=1.0$ |
> | ------------------- | :-----------: | :-----------: | :----------: |
> | $\text{Mixup}$ [1]  |    $9.82$     |    $10.13$    |   $10.21$    |
> | $\text{CutMix}$ [2] |    $10.29$    |    $10.18$    |   $10.43$    |
> |                     |  $8\times8$   | $12\times12$  | $16\times16$ |
> | $\text{Cutout}$ [3] |    $9.69$     |    $9.83$     |   $10.25$    |
>
> | $\text{Methods}$               | $\text{FID}$ |
> | ------------------------------ | :----------: |
> | $\text{EDM-DDPM++ (baseline)}$ |   $10.02$    |
> | $\text{WILD-Diffusion (Ours)}$ |  **$8.57$**  |
>
> [1] Zhang et al., "mixup: Beyond empirical risk minimization." ICLR 2018
>
> [2] Yun et al., "Cutmix: Regularization strategy to train strong classifiers with localizable features." ICCV 2019
>
> [3] DeVries & Taylor., "Improved regularization of convolutional neural networks with cutout." arXiv 2017
>
> [4] Zhang et al., "Training Diffusion-based Generative Models with Limited Data." ICML 2025
>
> **W4: Extension to other domain tasks. Experiments are primarily conducted on ... data-scarce adaptation would demonstrate stronger real-world relevance.**
>
> **A:** We thank the reviewer for the valuable suggestion. Although our primary focus is on unconditional generation (Table 1 for low-resolution and **Table 5 for high-resolution**), we agree that evaluating WILD-Diffusion in more diverse domains is important for demonstrating its broader applicability.
>
> To address this concern, we have **added new experiments** in the revised manuscript (Section 4.2 and Table 1), integrate WILD-Diffusion into a **class-conditional** diffusion model (i.e., Conditional EDM-DDPM++). These results confirm that our method consistently improves data-efficient performance beyond the unconditional setting.
>
> Table 3: Class-conditional generation results on CIFAR-10 ($32\times32$) under varying data availability (20%, 50%, 100%).
>
> | $\text{Methods}$                 | 20% $\text{data}$ | 50%$~\text{data}$ | 100%$~\text{data}$ |
> | -------------------------------- | :----------------: | :----------------: | :-----------------: |
> | $\text{EDM-DDPM++ (cond.)}$      |      $12.33$       |       $6.03$       |       $1.79$        |
> | $\text{+WILD-Diffusion (cond.)}$ | 10.89 (-11.68%) | 5.37 (-10.95%)  |  1.71 (-4.47%)   |
>
> In addition to the class-conditional experiments, we further evaluate WILD-Diffusion in a text-to-image personalization setting. We integrate WILD-Diffusion into a DreamBooth [1] fine-tuning method and conduct few-shot adaptation experiments. As summarized in Table 4, WILD-Diffusion consistently reduces overfitting (lower PRES) and improves subject fidelity (higher DINO and CLIP-I) under limited data. These results suggest that our plug-and-play framework can extend to text-conditioned large-scale diffusion models.
>
> Table 4:  Text-to-image personalization results on the DreamBooth dataset.
>
> | $\text{Methods}$                  | $\text{PRES} \downarrow$ | $\text{DINO} \uparrow$ | $\text{CLIP-I} \uparrow$ |
> | --------------------------------- | :----------------------: | :--------------------: | :----------------------: |
> | $\text{DreamBooth (Imagen)}$      |         $0.493$          |        $0.684$         |         $0.815$          |
> | $\text{+WILD-Diffusion (Imagen)}$ |         $0.478$          |        $0.696$         |         $0.823$          |
>
> [1] Ruiz et al., "DreamBooth: Fine Tuning Text-to-Image Diffusion Models for Subject-Driven." CVPR 2023

---

> ### Author Response · Authors · 2025-11-26
>
> Dear Reviewer `vXtY`,
>
> Thank you for taking the time to review our paper and provide valuable feedback. As the discussion phase is nearing its conclusion, we would like to confirm if our responses from a few days ago have effectively addressed your concerns. If you have any additional comments, we will do our best to address them.
>
> Best regards,
>
> The authors of Submission **9149**.

---

> > ### Comment · Reviewer_vXtY · 2025-11-27
> >
> > Thank you for the detailed rebuttal, and I have gone through all the newly added experimental results. The new experiments and the discussion addressed most of my main concerns. Overall, I believe the paper has been strengthened via rebuttal, and I am thus updating my score from 4 to 6.

---

> > > ### Author Response · Authors · 2025-11-27
> > >
> > > We thank the reviewer again for the positive and encouraging feedback.

---

### Author Response · Authors · 2025-11-21
**General Response**

We sincerely thank all the anonymous reviewers for their valuable and insightful comments. We have added some new content to our revised paper according to the comments and questions. The changes are summarized below and marked in ***Blue*** in the paper.

Revisions to the paper

- *page 9, Lines 442-444;* We have added a note clarifying that Patch Diffusion and DeepCache are included to demonstrate the plug-and-play nature of WILD-Diffusion (for **W2** of reviewer `vXtY`).
- *page 9, Lines 462, 464-465;* We have added two experimental results in Table 1 (for **W4** of reviewer `vXtY`; for **W1** of reviewer `rEDT`).
- *page 10, Lines 510-512;* We have added the results of two pre-trained models (StyleGAN-v2 and EDM-NCSN++) to Table 2 (for **W1** of reviewer `hDug`).
- *page 3, 19, Lines 129-133, 982-987;* We have added a reference [1] and a discussion to clarify the relevance of our work to it (for **W1** and **Q1** of reviewer `rEDT`).
- *page 24, Lines 1294, 1325-1339;* We have added Figure 5 to separately visualize the off-distribution samples produced by our bi-level update (for **Q4** of reviewer `rEDT`).
- *page 25, Lines 1345, 1574-1579;* We have added two datasets (CelebA-HQ and LSUN-Cat) to evaluate the generation performance of our method in Table 6 (for **W3** of reviewer `rEDT`).
- *page 30, Lines 1613-1630;* We have added a new Appendix C.5 to assess whether our method generalizes to Text-to-Image diffusion models (for **W4** of reviewer `vXtY`; for **Q3** of reviewer `hDug`; for **Q4** of reviewer `GEiA`)
- *page 31, Lines 1656-1659;* We added a clarification on the hyperparameter settings of Mixup, CutMix, and Cutout in our experiments (for **W3** of reviewer `vXtY`).
- *page 32, Lines 1690-1727;* We have added Table 10 and Figure 12 to illustrate the computational overhead of our method (for **W1** of reviewer `vXtY`; for **Q1** of reviewer `hDug`; for **Q3** of reviewer `GEiA`).
- *page 7;* We moved Algorithm 1 from Appendix A to Section 3.1.

[1] Wang et al., "Improved Diffusion-based Generative Model with Better Adversarial Robustness." ICLR 2025

---

### Author Response · Authors · 2025-12-03
**General Response to PCs, SACs, ACs, and Reviewers**

Dear PCs, SACs, ACs, and Reviewers,

We sincerely appreciate your diligent efforts and insightful feedback on our submission. Your comments have greatly helped us to improve the quality and clarity of our manuscript. To assist the newly assigned AC and help reduce the workload, we provide below a summary of the key points from the reviews and the reviewer-author discussions.

We are pleased that the reviewers recognized several **strengths** of our work, including:

- The importance and practical relevance of the addressed problem (Reviewers `vXtY`, `GEiA`).
- A solid theoretical foundation and technically sound methodology (Reviewers `vXtY`, `hDug`, `GEiA`, `rEDT`).
- The comprehensiveness of our experimental evaluations, especially in limited data settings (Reviewers `vXtY`, `GEiA`).
- High practicality and plug-and-play applicability of the method (Reviewers `hDug`,  `GEiA`, `rEDT`).
- Clear writing and logically structured presentation (Reviewers  `vXtY`, `rEDT`).

Following the valuable suggestions from the reviewers, we have provided detailed responses to all comments in the rebuttal. In particular, we have made the following major improvements:

- We have added the absolute measurements to assess the **computational overhead** of our method (Reviewers `vXtY`, `hDug`, `GEiA`).
- We have added two baseline methods (AT-Diff and A$^3$FT) and four datasets (ImageNet, Pokemon, MetFaces, and LSUN-Cat) to strengthen the evaluation of our method (Reviewers `hDug`, `rEDT`).
- We extended our WILD-Diffusion framework to more practical and diverse settings such as **class-conditional** generation and **text-to-image** personalization (Reviewers `vXtY`, `hDug`, `GEiA`).
- The ablation studies are included to demonstrate the effectiveness of the proposed WILD-Diffusion (Reviewers `vXtY`, `GEiA`).
- We explained that the methods A$^3$FT, DeepCache and Patch Diffusion are **orthogonal** to our method WILD-Diffusion. Namely, our method is plug-and-play and can be combined with them (Reviewers `vXtY`, `hDug`).
- We clarified that the dual surrogate does not introduce mis-specification of the worst-case set, and that Lipschitz/log-Sobolev–style assumptions are standard (Reviewer `GEiA`).
- We explained that the worst-case samples are usefully hard rather than toxic, from both the methodological and empirical perspectives  (Reviewer `GEiA`).
- We clarified the differences between our method and the method from Wang et al. (2025) [1] (Reviewer `rEDT`).
- We have added a figure that separately visualizes the samples generated by WILD-Diffusion (Reviewer `rEDT`).

[1] Wang et al., "Improved Diffusion-based Generative Model with Better Adversarial Robustness." ICLR 2025

------

## **Feedbacks from reviewers for our revision.**
Three reviewers acknowledged that their concerns had been satisfactorily addressed, and two of them raised their scores accordingly. **Reviewers `vXtY` and `rEDT`** increased their scores from **4 to 6**. **Reviewer `hDug`** maintained the score of **6**, confirming that the key concerns had been resolved. Probably due to the emergency shutdown of rebuttal discussion, **Reviewer `GEiA`** did not provide a response before that.

As a result, the score distribution before the rollback was: **6, 6, 6, and 6**, and these acknowledgments were explicitly documented in the discussion.

------

Above, we have faithfully summarized all reviewer comments and our corresponding responses, hoping that this will assist the newly assigned AC in evaluating our work.

We are deeply grateful to the reviewers, ACs, SACs and PCs for their time, insightful feedback, and dedicated efforts throughout this process. We believe that this review round has greatly improved the quality of our manuscript.

Best regards,

The authors of Submission **9149**.

---

### Meta-Review · Area_Chair_AvTC · 2025-12-29

**Summary:**

This paper proposes a training method for diffusion models under limited data inspired by the Wasserstein Distributionally Robust Optimization.

Reviewers commented positively on the important problem being addressed, the theory‑grounded and applicable method, the convincing experiments, and the writing and presentation quality

Reviewers raised questions about computational overhead, the selection and tuning of baselines, the generalisation ability of the method, dependence on assumptions, and the potential for more experiments. The concerns have been mostly addressed.

Overall, according to reviewers’ recommendations, the strengths outweigh the weaknesses.

**Reviewer Concerns:**

For Reviewer vXtY, the weaknesses have been well addressed.

For Reviewer hDug, the weaknesses have been well addressed.

For Reviewer GEiA, the weaknesses have been responded.

For Reviewer rEDT, the weaknesses have been largely addressed.

**Reviewer Scores:**

For Reviewer vXtY, the score has increased.

For Reviewer hDug, the original positive score has been re-confirmed.

For Reviewer GEiA, the score is likely to be the same or increased.

For Reviewer rEDT, the score has increased.

---

### Decision · Program_Chairs · 2026-01-26

Accept (Poster)